# Novel fast pathogen diagnosis method for severe pneumonia patients in the intensive care unit: randomized clinical trial

Yan Wang[1†], Xiaohui Liang[2†], Yuqian Jiang[2], Danjiang Dong[1], Cong Zhang[2], Tianqiang Song[2], Ming Chen[1], Yong You[1], Han Liu[3], Min Ge[4], Haibin Dai[5], Fengchan Xi[6], Wanqing Zhou[7], Jian-Qun Chen[2], Qiang Wang[2]*, Qihan Chen[2,8]*, Wenkui Yu[1,8]*

[1]Department of Critical Care Medicine, Nanjing Drum Tower Hospital, The Affiliated Hospital of Nanjing University Medical School, Nanjing, China; [2]The State Key Laboratory of Pharmaceutical Biotechnology, School of Life Sciences, Nanjing University, Nanjing, China; [3]Department of Critical Care Medicine, Nanjing First Hospital, Nanjing Medical University, Nanjing, China; [4]Department of Cardiothoracic Surgery Intensive Care Unit, Nanjing Drum Tower Hospital, The Affiliated Hospital of Nanjing University Medical School, Nanjing, China; [5]Department of Neurosurgery Intensive Care Unit, Nanjing Drum Tower Hospital, The Affiliated Hospital of Nanjing University Medical School, Nanjing, China; [6]Research Institute of General Surgery, Affiliated Jinling Hospital, Medical School of Nanjing University, Nanjing, China; [7]Department of Laboratory Medicine, Nanjing Drum Tower Hospital, The Affiliated Hospital of Nanjing University Medical School, Nanjing, China; [8]Medical School of Nanjing University, Nanjing, China

*For correspondence:
wangq@nju.edu.cn (QW);
chenqihan@nju.edu.cn (QC);
yudrnj@nju.edu.cn (WY)

[†]These authors contributed equally to this work

Competing interest: The authors declare that no competing interests exist.

## Abstract

**Background:** Severe pneumonia is one of the common acute diseases caused by pathogenic microorganism infection, especially by pathogenic bacteria, leading to sepsis with a high morbidity and mortality rate. However, the existing bacteria cultivation method cannot satisfy current clinical needs requiring rapid identification of bacteria strain for antibiotic selection. Therefore, developing a sensitive liquid biopsy system demonstrates the enormous value of detecting pathogenic bacterium species in pneumonia patients.

**Methods:** In this study, we developed a tool named Species-Specific Bacterial Detector (SSBD, pronounce as 'speed') for detecting selected bacterium. Newly designed diagnostic tools combining specific DNA-tag screened by our algorithm and CRISPR/Cas12a, which were first tested in the lab to confirm the accuracy, followed by validating its specificity and sensitivity via applying on bronchoalveolar lavage fluid (BALF) from pneumonia patients. In the validation I stage, we compared the SSBD results with traditional cultivation results. In the validation II stage, a randomized and controlled clinical trial was completed at the ICU of Nanjing Drum Tower Hospital to evaluate the benefit SSBD brought to the treatment.

**Results:** In the validation stage I, 77 BALF samples were tested, and SSBD could identify designated organisms in 4 hr with almost 100% sensitivity and over 87% specific rate. In validation stage II, the SSBD results were obtained in 4 hr, leading to better APACHE II scores (p=0.0035, ANOVA test). Based on the results acquired by SSBD, cultivation results could deviate from the real pathogenic

situation with polymicrobial infections. In addition, nosocomial infections were found widely in ICU, which should deserve more attention.

**Conclusions:** SSBD was confirmed to be a powerful tool for severe pneumonia diagnosis in ICU with high accuracy.

**Funding:** National Natural Science Foundation of China. The National Key Scientific Instrument and Equipment Development Project. Project number: 81927808.

**Clinical trial number:** This study was registered at https://clinicaltrials.gov/ (NCT04178382).

## Editor's evaluation

Current culture-based, gold standard methods used for diagnosing the cause of sepsis provide results in 48-96 hours slowing antibiotic treatment initiation and leading to poor patient recovery. This work provides a new tool for identifying sepsis- and pneumonia-causing pathogens in less than 4 hours with species-specificity with the hope that the fast turnaround time leads to early treatment and improved clinical outcomes. Using an optimized PCR+CRISPR-Cas12a DNA detection method, the assay demonstrates good analytical sensitivity and specificity for 10 common bacterial pathogens that cause pneumonia. The method is validated in a clinical cohort and the clinical benefit is analyzed using a second cohort which is an intervention study used to guide clinicians on treatment choice.

## Introduction

Sepsis is associated with high morbidity and mortality (*Singer et al., 2016*). Adequate antibiotic therapy in time could decrease mortality and reduce the length of stay in ICU for patients with sepsis or septic shock (*Ferrer et al., 2018*; *Kumar et al., 2006*; *Pulia and Redwood, 2020*; *Seymour et al., 2017*). As reported in the previous study, the mortality rate of patients increased approximately 7.6% for every hour delayed (*Kumar et al., 2006*). Therefore, rapid diagnosis of pathogenic microorganisms is crucial for shortening the time of empirical antibiotic therapy and improving the prognosis of patients with sepsis.

Conventional culture test (CCT) is the most commonly used and golden standard identification method of pathogenic microorganisms in most countries. However, it showed two critical limitations: long time-consuming (2–5 days) and low sensitivity (30–50%), which limited the application of this method in the ICU (*Abd El-Aziz et al., 2021*; *Zhou et al., 2014*). To overcome this bottleneck, several new tools were developed and showed significant improvement in time consumption and accuracy. Recently, next-generation sequencing (NGS) technology was applied to acquire the entire information of microorganisms and demonstrated great ability in diagnosing rare pathogens. However, the whole process still needs at least 2 days for the full diagnostic report with high cost (*Chen et al., 2020*; *Wang et al., 2020*). On the other hand, NGS provided too much information about microorganisms but only semi-quantification of pathogens, which was hard for most clinical doctors to extract the most important information to determine antibiotic usage. Other new emerging detection techniques designed by BioFire and Curetis are much superior in detection time than these above. However, its original principle was based on nucleotide diversity of conserved genes among species, which could not satisfy the application in the ICU due to potential false-positive results (*Edin et al., 2020*; *Jamal et al., 2014*; *Trotter et al., 2019*). Therefore, a unique diagnosis tool aimed at faster and more accurate pathogen identification in the ICU was still a great challenge.

In this study, we aimed to design a simple and convenient diagnosis tool for sepsis patients in the ICU, which covered the most common pathogenic bacteria and completed the detection process in the shortest possible time with low cost and minimum instrument requirements. A clinical trial with two stages was applied to evaluate the accuracy of the tool and the clinical benefits.

## Materials and methods

### Study design

The full study design was shown in *Figure 1*. In the discovery stage, we screened species-specific DNA tags of 10 epidemic pathogenic bacteria in the ICU. In the training stage, we optimized reaction conditions and sample preparation process, including detection concentration limitation, DNA purification, and incubation time of the CRISPR/Cas12a reaction. The finalized experiment operating procedure of SSBD was used in the subsequent stages (detailed protocol was shown in Appendix 1).

In validation stage I, 77 specimens of bronchoalveolar lavage fluid (BALF) directly acquired from patients in ICU were finally detected by SSBD to confirm the specificity and sensitivity of SSBD compared to CCT results. Based on clinical needs, some of the samples were diagnosed by NGS technology in third party commercial company, which provided additional information for reference.

After the stability and accuracy of SSBD were thoroughly evaluated, the validation stage II, a preliminary clinical intervention experiment, was launched to verify the clinical application of SSBD.

### Screening species-specific DNA tags

We designed a process to find the species-specific DNA tags according to the basic principle, intraspecies-conserved and interspecies-specific sequences (illustrated in *Figure 2A*). A total of 1791 high-quality genomes of 232 microorganism species from the public databases were included in the screening process. To accelerate the screening process, we developed a linear comparison algorithm instead of comparing every two genomes, which could save more than 90% of calculation time cost (*Appendix 1—figure 1*). According to the epidemiological data by previous retrospective study (*Zhou et al., 2014*) and 2017 data in ICU of Nanjing Drum Tower Hospital (*Appendix 1—figure 2*), 10 species of bacteria covered 76% sepsis pathogenic bacteria and therefore were selected as targets for subsequent detecting process, including *Acinetobacter baumannii* (*A. baumannii*), *Escherichia coli* (*E. coli*), *Klebsiella pneumoniae* (*K. pneumoniae*), *Pseudomonas aeruginosa* (*P. aeruginosa*), *Stenotrophomonas maltophilia* (*S. maltophilia*), *Staphylococcus aureus* (*S. aureus*), *Staphylococcus epidermidis* (*S. epidermidis*), *Staphylococcus capitis* (*S. capitis*), *Enterococcus faecalis* (*E. faecalis*) and *Enterococcus faecium* (*E. faecium*). Then we designed different DNA primers targeting selected species-specific DNA tags from each species (*Appendix 1—tables 1 and 2*).

To evaluate our primers' specificity in identifying species, we chose *S. aureus* and *S. epidermidis* from the same genus as our cross-validated target species. We extracted DNA sequences of the *S. aureus* and the *S. epidermidis* amplified by primers used in FilmArray Pneumonia Panel developed by BioFire and in our protocol, which were acquired from NCBI Reference Prokaryotic Representative genomes. We then aligned *S. aureus*-specific DNA sequences with the representative genome of *S. epidermidis* using blast to search the most similar DNA sequences. In the FilmArray Pneumonia Panel, DNA amplified sequences from the *S. aureus* and the *S. epidermidis* were aligned to each other (two different gene regions, *rpoB* and *gyrB*, were used to separate two species).

### Patients

Patients admitted to ICUs and diagnosed with severe pneumonia were recruited from Aug 27, 2019. The recruit criteria for patients were: (1) age ≥18 years; (2) had artificial airway and expected to retain for more than 48 hr; (3) clinically diagnosed as pneumonia, and the microbiology of etiology was unclear; (4) signed informed consent; (5) the expected length of staying in ICU was more than 3 days. According to previous mortality acquired from the adequate anti-infective group, the sample size calculation (two-group rate) for patients was done, and a sample size of 73 patients in each group was needed. The enrolled participants formed a consecutive and convenient series, who were randomly assigned to experimental or control groups as described in the appendix 1.

### Clinical outcomes

BALFs were obtained from all the patients from 2 groups on day 1, day 3–5, and day 7–10 after recruitment and were sent directly to the hospital diagnostic microbiology laboratory for CCT and susceptibility testing. CCT results were obtained in strict accordance with international ERS/ESICM/ESCMID/ALAT guidelines for the management of hospital-acquired pneumonia and ventilator-associated pneumonia, which is currently the guideline for clinical gold standard. BALFs from patients of the experiment group were also sent for SSBD tests immediately after sampling. The time from sample

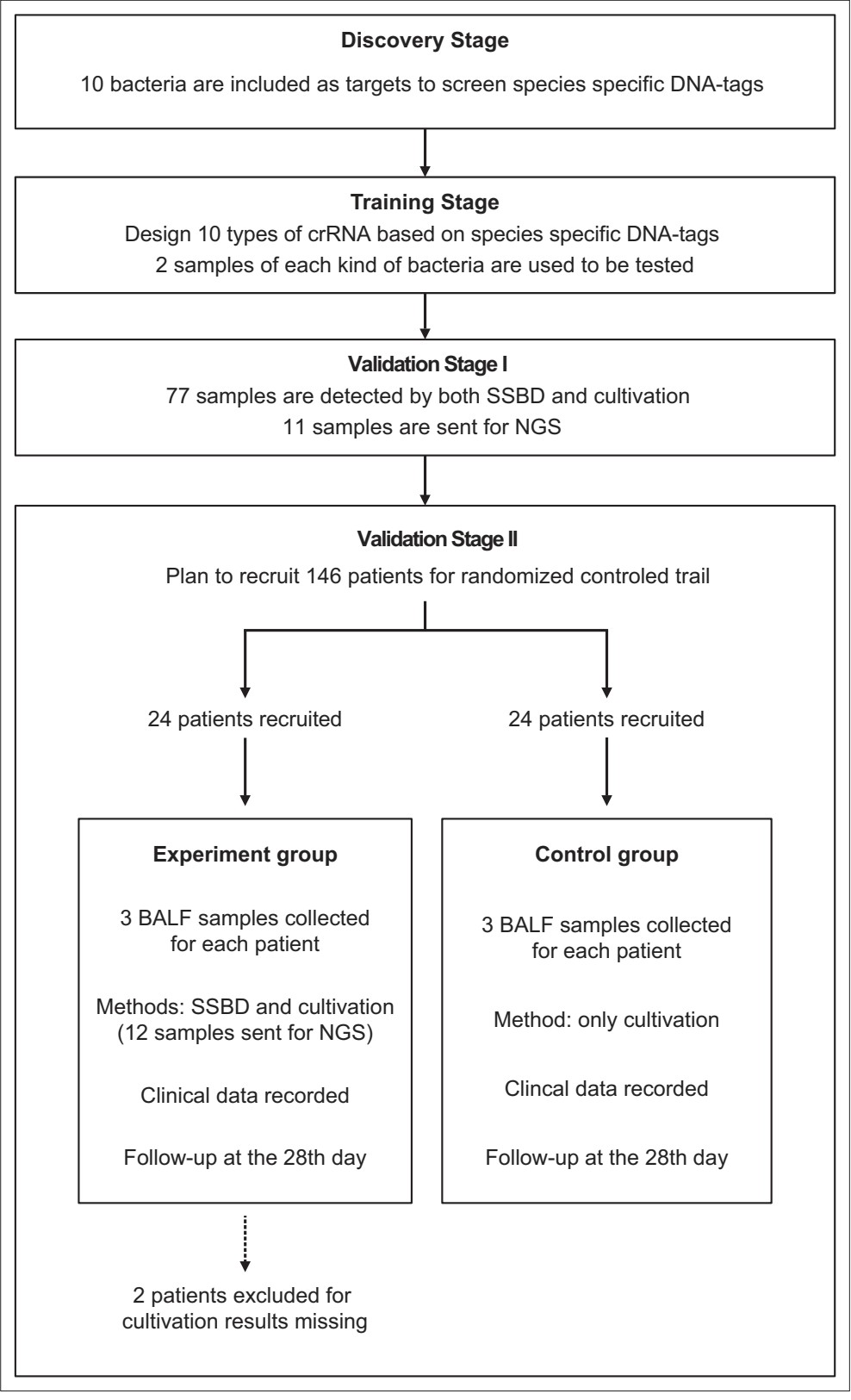

**Figure 1.** Study design. This study contained four stages: discovery stage, training stage, validation stage I and validation stage II. All patients were from the Department of Critical Care Medicine, Nanjing Drum Tower Hospital. Patients were randomly divided into two groups for the clinical trial.

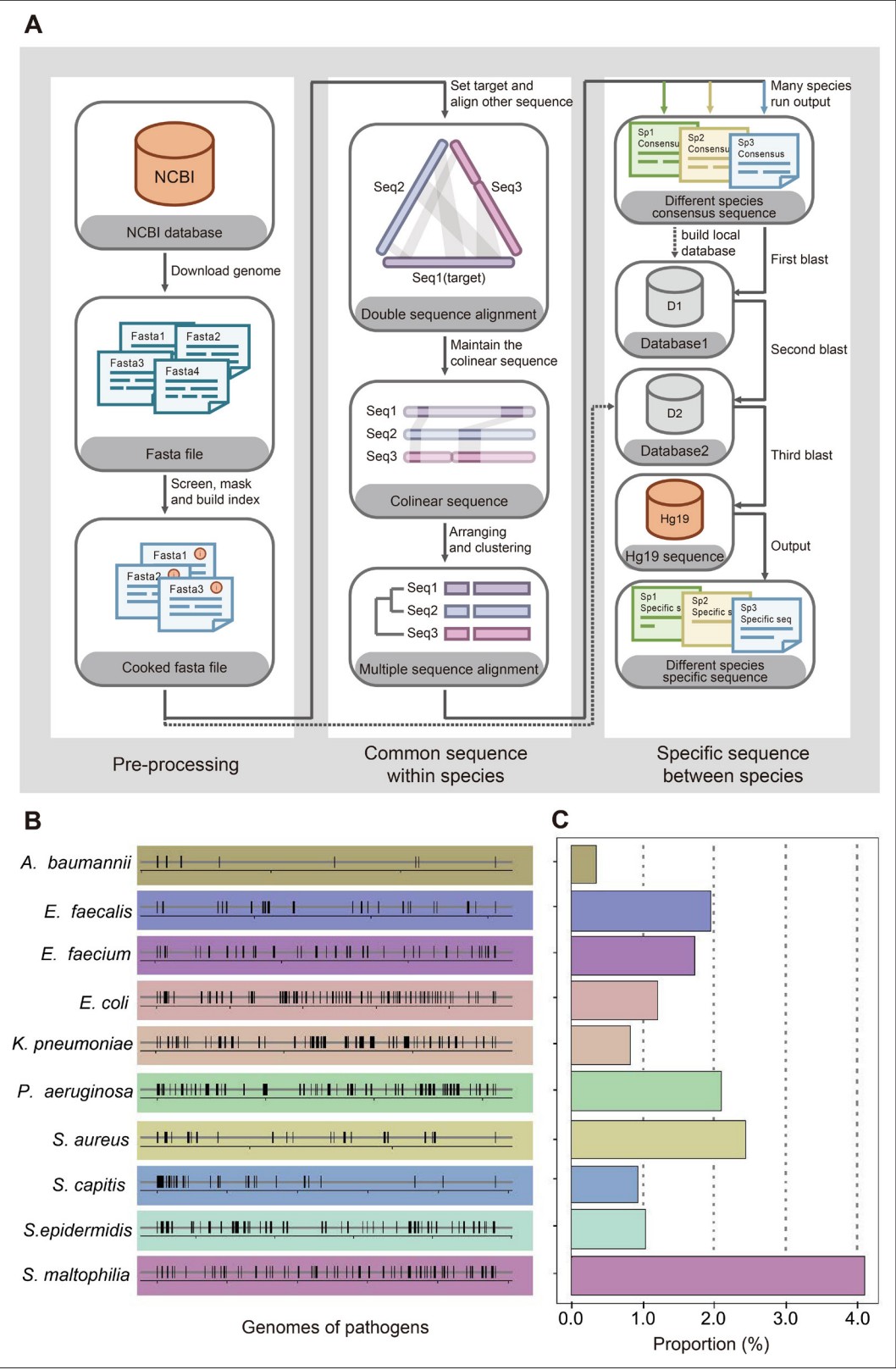

**Figure 2.** Screening workflow and statistics of species-specific DNA tags. (**A**) Schematic diagram of screening species-specific DNA tags. (**B**) Genomic distribution of species-specific DNA tags in 10 bacteria. (**C**) Genomic proportion of species-specific DNA tags in 10 bacteria.

acquisition to feedback of results was defined as the turnaround time, as well as being estimated and compared between SSBD and CCT. Other clinical records included blood routine tests, CRP and PCT examinations.

All enrolled patients received primary empirical antibiotic therapy. Once the SSBD results of the patients in the experiment group were obtained, the decisions about whether antibiotics were adjusted or not were made by two senior doctors according to the SSBD results and other clinical information. While in the control group, adjustment depended on conventional culture results and clinical data. Patient demographics and other vital clinical parameters were recorded. Acute Physiology and Chronic Health Evaluation II (APACHE II) scores and Sequential Organ Failure Assessment (SOFA) scores were calculated and recorded for patients on days 1, 3, 7, 10, and 14 to assess their disease severity and organ function.

### Statistical analysis

The number of improved patients on different clinical indicators of different days was calculated and tested by Fisher's exact test. APACHE II scores and SOFA scores were tested as a series by two-way ANOVA. Different clinical outcomes were tested by the Mann-Whitney test.

### Funding support

This study was funded by National Natural Science Foundation of China. The National Key Scientific Instrument and Equipment Development Project. Project number: 81927808. The funders were not involved in the initiation or design of this study, collection of samples, analysis and interpretation of data, writing of the paper, or the submission for publication. The study and researchers are independent of the funders.

## Results

### The identification of species-specific DNA fragments

The first step to identify pathogenic bacteria was to figure out the specific genome information of each species. Bacteria were quite similar between close-related species but sometimes quite different among different strains of one species due to fast evolution and horizontal gene transfer (*Dombrowski et al., 2020*; *Brito, 2021*; *Groussin et al., 2021*), which makes it hard to figure out great species-specific DNA fragments. For example, two typical strains PAO1 and PAO7 of *P. aeruginosa* (NCBI representative genome database) demonstrate less than 94% nucleotide identity, while *E. coli* and *Shigella sonnei* both belong to Enterobateriaceae and their representative genomes share more than 98% nucleotide identity. Therefore, the widely used method to identify bacteria with conserved genes may not be a good choice (*Maslunka et al., 2015*; *Liu et al., 2021*). We developed an innovative algorithm and designed a workflow to figure out the best DNA tag for each species for diagnostic application based on 1791 microbe genomes from 232 species (*Figure 2A*). The details could be found in Appendix 1.

We started from 10 common bacteria contributing to sepsis infection as the initial panel according to local epidemic data from ICU of Drum Tower hospital and previous studies about pathogens in ICU (*Appendix 1—figure 2*; *De Pascale et al., 2012*; *Sakr et al., 2018*). To our surprise, bacteria-specific DNA sequences showed a random distribution and turned out to be only 0.3–4.1% in the whole genomes of 10 bacteria (*Figure 2B and C*). Considering the application scenario of ICU with only basic instruments, PCR +CRISPR/Cas12a system was chosen for the following detection. Based on the identified species-specific DNA fragments, related primers and crRNAs (CRISPR RNA) were designed according to each species (*Appendix 1—tables 1 and 2*).

### The establishment of species-specific bacteria detection tool

Briefly, CRISPR/Cas12a with designed crRNA could be activated by its target, which could be told by whether the reporter probe was cleaved and demonstrated signal as previously reported (*Chen et al., 2018*).

To optimize the working conditions of the detection tool in ICU, multiple experiments were applied to optimize the sample preparation and detection process. With the gradient concentration of DNA templates, we confirmed that the lowest detection limit was $10^{-15}$ M with PCR amplification and $10^{-8}$ M

without amplification step (*Figure 3B*), which was consistent with previous studies (*Gootenberg et al., 2018*). In addition, 30 min' incubation of CRISPR/Cas12a with PCR products was enough to demonstrate signals (*Figure 3B*). An additional purification step right after PCR amplification appeared unnecessary to acquire the positive result but helpful for weaker signal (*Appendix 1—figure 3A*). In addition, the comparison of CRISPR/Cas incubation duration confirmed that fluorescence value showed a significant difference from 5 min and reached its maximum after 30 min compared to the negative control (*Appendix 1—figure 3B*).

To confirm the primary behavior of SSBD, two clinical strains separated from different patients for each of 10 selected bacteria species were collected and tested by SSBD as the positive control, which showed clear positive results (*Appendix 1—figure 3C*). To further confirm the specificity of SSBD, each bacteria strain was tested by 10 SSBD test panels targeting different bacteria. Compared to negative control, only SSBD targeting the tested bacteria showed a positive result, which confirmed its high specificity (*Figure 3C*).

Putting these results together, a standard operating procedure was finally established for the following validation stages (*Figure 3A*), which was capable of providing the information about the ten most common pathogenic bacteria in ICU. Since this method was a quite fast and species-specific bacteria detection tool, we named it SSBD.

## The accuracy and clinical benefits of SSBD

We started our study with validation stage I, which was a non-intervention study with 77 samples of BALF extracted from patients. Samples were detected both by SSBD and CCT, and the results were compared (raw detection results were shown in *Appendix 1—table 3*).

Generally, 5 of 10 selected bacteria were detected by both tests, including *A. baumannii*, *K. pneumoniae*, *P. aeruginosa*, *S. aureus,* and *S. maltophilia*. SSBD could detect those five bacteria separately with 100% sensitivity and over 87% specificity, which were calculated by the results of CCT as golden standard (*Figure 4A*). The other five bacteria were detected by SSBD but not CCT, including *E. coli*, *S. epidermidis*, *S. capitis*, *E. faecalis,* and *E. faecium*. Among all samples, 11 of them were determined by patients to acquire results with NGS, which provided extra information to evaluate the results (*Appendix 1—table 4*). Based on the results, SSBD was highly consistent with NGS, which implied that SSBD might provide more accurate and complete pathogenic information than CCT in the selected panel.

Based on these accurate results, we started the validation stage II, which was an intervention study aiming to evaluate the clinical benefits of SSBD compared to the current diagnosis and treatment strategy in ICU. Although the study was paused due to the outbreak of SARS-CoV2, 22 patients were recruited into the experiment group and 24 patients into the control group. The baseline characteristics had no significant difference except ages (*Table 1*).

We finally got 57 BALF results tested by SSBD, which included 43 results that also had CCT results among them in the experiment group. While in the control group, we got 63 samples tested only by CCT. In the experiment group, 47 samples were positive among 57 samples tested by SSBD, while 28 samples were positive among 43 samples tested by CCT (raw detection results were shown in *Appendix 1—table 5*). In the control group, 41 samples showed positive among 63 samples. It was shown that SSBD could detect each bacterium with similar high sensitivity and specificity in validation stage II (*Figure 4B*). Consistent with the local epidemic data, the most frequent occurrence was *A. baumannii* (*Figure 4B*). Similar to stage I, 12 samples were determined by patients to test with NGS help us to draw the same conclusion that SSBD seemed to be better that CCT (*Appendix 1—table 6*).

To explore clinical benefits with the help of SSBD, effective antibiotic coverage rate, APACHE II scores and SOFA scores were calculated and compared to evaluate the rationalization of antibiotic therapy and patients' disease severity and organ function status in the two groups (*Figure 4C-E*). Effective antibiotic coverage rates for each test were significantly higher in the experimental group than those in the control group in three tests (*Figure 4C*). The definition of antibiotic coverage and the original calculation results were shown in *Appendix 1—figure 4A and B*. APACHE II scores were significantly lower in the experimental group than those in the control group after day 1 (p=0.0035, two-way ANOVA); the separation between two groups of patients increased progressively until day 14 (*Figure 4D*). SOFA scores showed no difference between the groups (p=0.8918, two-way ANOVA) (*Figure 4E*).

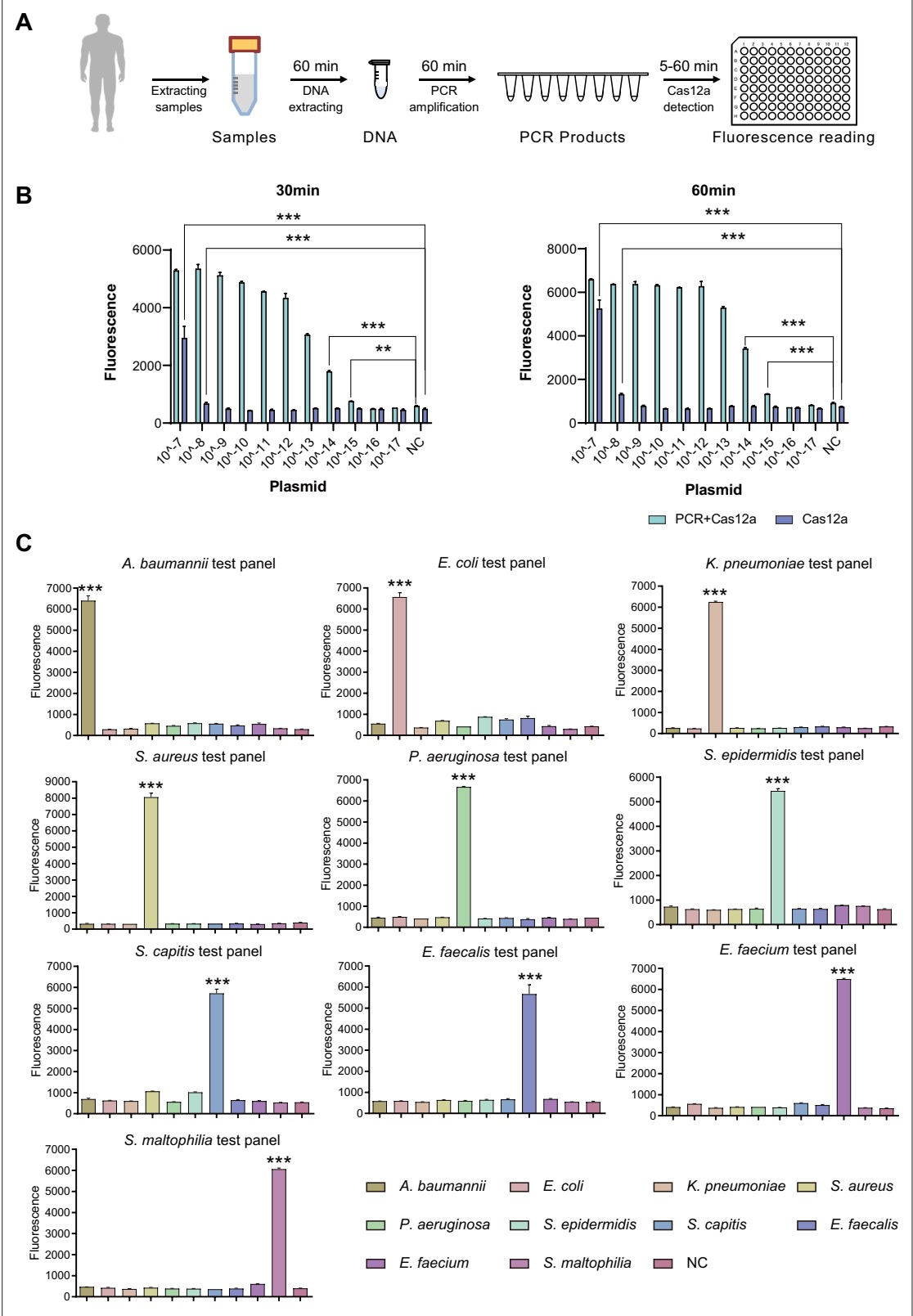

**Figure 3.** SSBD development and effectiveness validation. (**A**) SSBD workflow for clinical validation stages. (**B**) Cas12a and Cas12a-after-PCR detection of different concentrations and reaction times including 30 min (left) and 60 min (right). Blue bars indicated the Cas12a-after-PCR test. Brown bars indicated Cas12a test only. The concentration gradient of pGL3 plasmid from $10^{-17}$ M–$10^{-7}$M was established as the test group. NC stood for the fluorescence values of PCR products of using DEPC-H2O as input. Each group had three repeats. Error bars indicated mean ± SEM of fluorescence

*Figure 3 continued on next page*

*Figure 3 continued*

value. ** indicated p-value <0.01 and *** indicated p-value <0.001 of unpaired t-test. (**C**) SSBD results of 10 pathogenic bacteria. Every test panel for each of 10 bacteria was used to detect genome DNA samples of 10 bacteria by SSBD. NC stood for the fluorescence values of PCR products of using DEPC-H2O as input. Each group had three repeats. Error bars indicated mean ± SEM of fluorescence value. *** indicated p-value <0.001 of unpaired t-test.

The online version of this article includes the following source data for figure 3:

**Source data 1.** The reaction condition test of Cas12a detection.

**Source data 2.** The cross-validation of 10 selected bacteria using SSBD.

## Polymicrobial infection and nosocomial events observed by SSBD

Based on the previous studies, CCT had defects in the evaluation of polymicrobial infection events due to the limitations of its technology (*Azevedo et al., 2017*). Therefore, we tried to evaluate whether SSBD demonstrated better performance with polymicrobial infection. Here, we defined situations of infection with more than one pathogenic microorganism as polymicrobial infection events to assess the performance based on the results of both methods. From the results, the detection rate of polymicrobial infection events by SSBD was 41.8% (55/134) in two validation stages, which was significantly higher than 11.7% (14/120) of CCT (*Figure 5A*). Polymicrobial infection events were compared among SSBD, CCT and NGS, which demonstrated high consistency of SSBD and NGS (*Figure 5B*).

Since both SSBD and NGS were based on target DNA, we wanted to confirm if some polymicrobial infection events were 'false positive' and caused by dead bacteria. Here, we showed patient B19 as an example, who received three times tests at days 1, 3, and 7 by both CCT and SSBD. Based on the results, *S. maltophilia* was detected as level II in test1 with SSBD but not CCT. Later on, *S. maltophilia* was detected by CCT in test2 as well with few days' development from level II to level III based on result of SSBD, which means SSBD discovered the true polymicrobial infection event earlier than CCT (*Figure 5C*). From the aspect of pathogen species participated in polymicrobial infection events, both methods demonstrated similar results with *A. baumannii*, *S. maltophilia*, *P. aeruginosa*, *K. pneumoniae* and *S. aureus* in top 5 (*Figure 5D*), which were consistent with the frequency of pathogens in ICU (*De Pascale et al., 2012*).

Hospital infections, also known as nosocomial infections, are an important factor in the incidence rate and mortality of ICU patients with severe pneumonia (*Zaragoza et al., 2020*). Since CCT has a long delay in clinical feedback of pathogenic results, there is no effective monitoring method in clinical practice. Here, we tried to evaluate nosocomial infections based on the test results. We defined a case as a nosocomial infection event if a pathogenic bacterium was newly detected in the current time point but not before. For example, B17 (*K. pneumoniae* at test 2, *E. faecalis* at test 3) and B19 (*S. maltophilia* at test 2, *K. pneumoniae* at test 3) patients were discovered as nosocomial infection cases for SSBD and CCT (*Figure 5E and F*). Based on the results of SSBD, 47.6% (10/21) of patients had nosocomial infections at the test 2, and 28.6% (4/14) of patients had nosocomial infections at the test 3. Similarly, 40% (4/10) of patients were identified as nosocomial infections by CCT at test 2, and 27.3% (3/11) of patients were identified as nosocomial infections at test 3 (*Figure 5G*).

## Discussion

In this study, we developed a rapid bacteria detection technique based on CRISPR/Cas12a using species-specific DNA tags and detected common bacteria taken from pneumonia in 4 hr with 100% sensitivity and over 87% specificity in the validation stage I. Currently, there are already some market-oriented detection technologies for pneumonia patients, such as FilmArray Pneumonia Panel by BioFire and Curetis Unyvero system, which also could detect microorganisms in several hours (*Trotter et al., 2019*). However, based on the information in their product instruction, false positive results were widely seen in close relative species. Such problem may due to the marker selection strategy. For example, sequences used by FilmArray Pneumonia Panel from two gene regions had highly similar DNA sequences in the *S. epidermidis* representative genome (E-value=5e-40, *rpoB*; E-value=8e-39, *gyrB*), which could interfere with pathogen identification between species from the same genus. It was ideal for early and rapid screening of infectious diseases but was not applicable in the ICU, considering the complexity and urgency of infection events within the ICU. We have adopted a completely

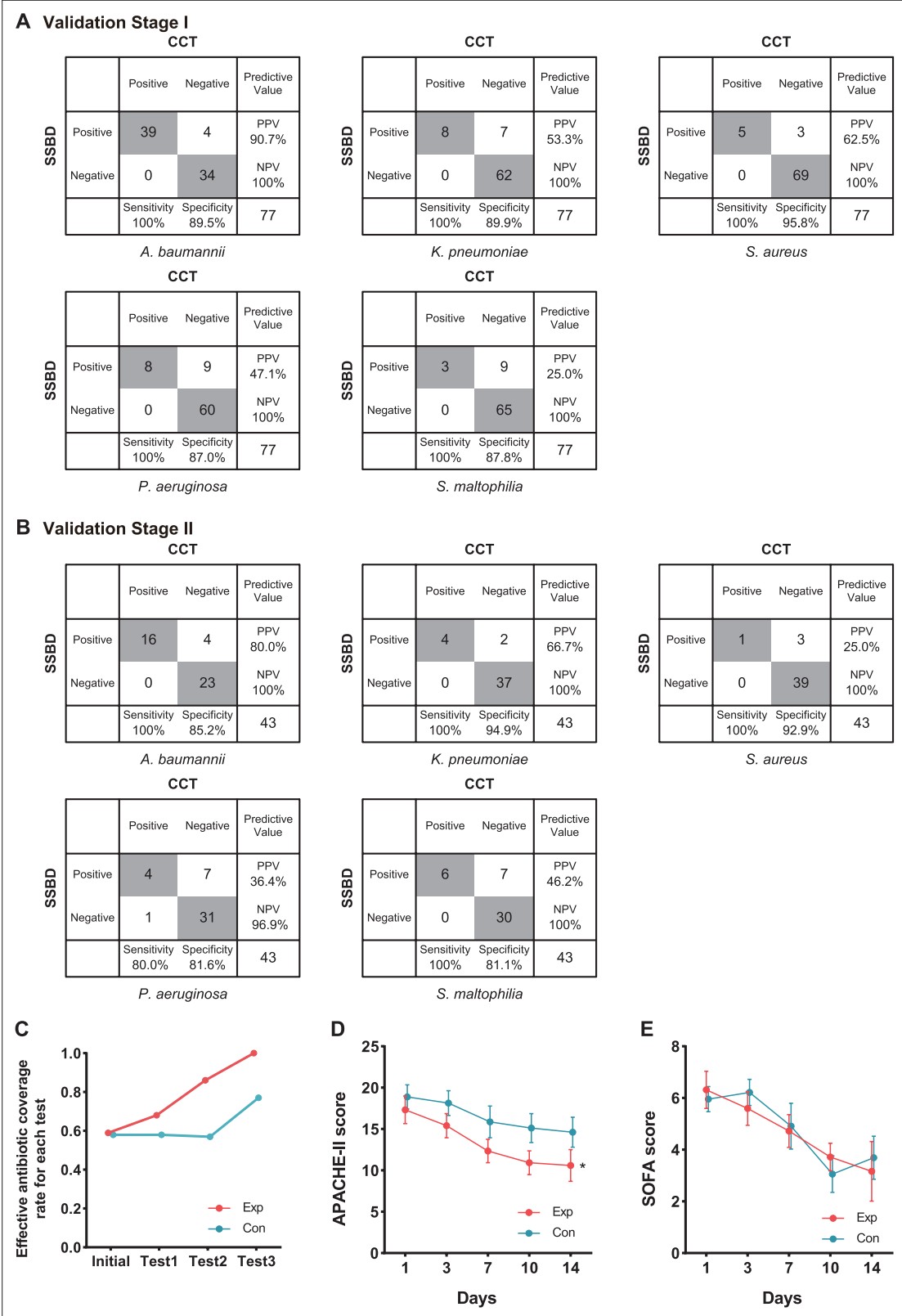

**Figure 4.** Statistical analysis of test results and clinical outcomes in the two validation stages. (**A**) Cross-tables for 5 of 10 bacteria by both SSBD and CCT in the validation stage I. (**B**) Cross-tables for 5 of 10 bacteria by both SSBD and CCT in the validation stage II. (**C**) Antibiotics coverage rate of each test in the two groups. Exp meant the experimental group, and Con meant the control group. Test 1: Day 1. Test 2: Day 3–5. Test 3: Day 7+. Raw antibiotics coverage results of each patient were available in *Appendix 1—figure 4B*. Detailed judging guidelines were shown in Appendix 1. (**D**

*Figure 4 continued on next page*

*Figure 4 continued*

**and E**) Line charts for APACHE II and SOFA scores, respectively. Error bars indicated mean ± SEM of scores of all the recorded patients. * indicated a significant difference between the two groups using two-way ANOVA.

different strategy from the existing methods, getting specific gene regions from species for further test using our developed bioinformatics workflow and algorithm. It was shown that our sequences used for *S. aureus* diagnosis had no similar fragments in *S. epidermidis*, which avoided distinguishing different species by gene diversity. It was likely to get the species-specific DNA tags from such amount genomes when aligned bacterial genomes with each other but consuming computational cost. We optimized calculation processes by rescheduling steps and then made it possible for us to acquire species-specific DNA regions after shortening time to a range bearable.

NGS technology is useful in species identification and also shows its advantages in clinical diagnosis. It is valuable to detect uncommon pathogens because of its unique capability in detecting multiple agents across the full microbial spectrum contributing to disease and has already been developed as a new detection platform (*Wang et al., 2020*). However, in the majority of cases of common pathogens, redundant microorganism results were probably unhelpful to the anti-infection regimen. In addition, the high cost and relatively long turnaround time prevent its widespread application, especially in the ICU circumstance. Therefore, our SSBD method seemed more advantageous in time-consuming and information effectiveness than other mentioned methods, especially when we could quantify bacterial load based on fluorescence intensity for better antibiotic therapy strategy. CRISPR/Cas12a and qPCR are both quantitative methods, but CRISPR/Cas12a shows its robustness and lower equipment requirement, which satisfied our needs for most of the ICU. There are still several challenges in implementing POCT in developing countries, especially the qPCR/POCT system, which will be an alternative. Here, we listed a table to demonstrated the comparison among SSBD, CCT and NGS from main aspects (*Appendix 1—table 10*).

The results of SSBD demonstrated high sensitivity and specificity. However, we discovered several 'false positive' results compared to CCT, which might be caused by two reasons: (1) The low bacterial load of the patient sample was probably not enough or needed much longer time than expected to be cultivated. SSBD provided a lower threshold of detection ($10^{-15}$ M) than CCT, which could detect pathogens that even existed in trace amounts which unable to be cultivated. In our study, the fluorescence intensity obtained from SSBD was divided into three intervals (level I: $10^{-15}$-$10^{-14}$ M, level II: $10^{-14}$ M-$10^{-13}$ M, level III: over $10^{-13}$ M), representing the different strengths of bacteria (roughly equivalent to bacteria amounts according to our lowest detection thresholds, dividing details in SSBD diagnostic report of Appendix 1). All false-positive results were calculated on the count of species and strengths, mostly belonging to the level I or II (*Appendix 1—figure 5*). Considering most of those false positive samples were also validated by NGS technology, it suggested that some pathogens might be missed in the CCT results. (2) Cultivation could fail in detecting pathogens that failed in competitive growth environments. It was interesting to see that many patients were infected by more than one pathogen, which might cause potential competition between different pathogens in CCT process (*Appendix 1—table 9*). For example, *A. baumannii* was found to be the most competitive bacteria in cultivation, which may be due to its fastest growth rate. On the other hand, *P. aeruginosa* seemed to be relatively the weakest one among them, which was usually concealed in the cultivation with other species existing (sample A16, B19-3, B21-1, B21-2 after we excluded all samples with *A. baumannii* existing).

When evaluating the clinical benefit from SSBD, the quicker directed therapy adjustment for patients in the experiment group (Exp: 10.2±8.8 hours vs. Con: 96.0±35.1 hr, p<0.0001, Mann-Whitney test) could shorten the empirical anti-infection time and seemed to alleviate illness severity (APACHE II score) during the validation stage II with the help of the SSBD. As showed in *Appendix 1—table 8*, patients in experiment groups for example demonstrated significant better measures of temperature improvement at day 3, WBC improvement at day 7. It implied that appropriate antibiotic treatment guided by in-time pathogenic information would alleviate acute physiological illness. Nevertheless, at the endpoint, clinical outcomes showed no differences between the two groups, which may due to the insufficient patient numbers.

Despite the size in our intervention stage, there were still some aspects that have not been considered. (1) Resistance genes were not included in the study. Multi-drug resistant organisms (MDROs) prevailed in ICU (*Liu et al., 2020*; *Yang et al., 2020*), which might not improve the situation of patients

**Table 1.** Demographic and baseline characteristics of the patients in the validation stage II.

| | Experimental group (n=22) | Control group (n=24) | p value |
|---|---|---|---|
| Women | 9 (40.9%) | 11 (45.8%) | 0.774 |
| Men | 13 (59.1%) | 13 (54.2%) | 0.774 |
| Age, years (SD) | 58 (17.4) | 68 (9.5) | 0.015* |
| Patients' numbers of chronic comorbidities | | | |
| Hypertension | 9 (40.9%) | 17 (70.8%) | 0.073 |
| Coronary artery disease | 1 (4.5%) | 3 (12.5%) | 0.609 |
| Chronic pulmonary disease | 2 (9.1%) | 4 (16.7%) | 0.667 |
| Chronic kidney disease | 2 (9.1%) | 6 (25.0%) | 0.247 |
| Diabetes | 5 (22.7%) | 12 (50.0%) | 0.072 |
| Malignancy | 0 (0.0%) | 2 (8.3%) | 0.490 |
| Stroke | 3 (13.6%) | 8 (33.3%) | 0.171 |
| Immunodeficiency/immune suppressive therapy | 5 (22.7%) | 3 (12.5%) | 0.451 |
| Recent surgery | 4 (18.2%) | 3 (12.5%) | 0.694 |
| Hemodynamic support (using vasoactive drugs) | 7 (31.8%) | 7 (29.2%) | 1.000 |
| Norepinephrine ≤0.1 μg/(kg•min) | 2 | 3 | |
| Norepinephrine >0.1 μg/(kg•min) | 1 | 1 | |
| Dopamine ≤5 μg/(kg•min) | 3 | 2 | |
| Dopamine >5 μg/(kg•min) | 0 | 1 | |
| Dobutamine ≤5 μg/(kg•min) | 1 | 0 | |
| Dobutamine >5 μg/(kg•min) | 0 | 0 | |
| Status at randomization (D1) | | | |
| Temperature, °C | 38.4 (0.6) | 38.3 (0.7) | 0.345 |
| Coma | 6 (27.3%) | 6 (25.0%) | 1.000 |
| Systolic blood pressure, mmHg | 112.2 (19.3) | 121.6 (17.8) | 0.057 |
| Invasive mechanical ventilation | 20 (90.9%) | 24 (100.0%) | 0.223 |
| Renal replacement therapy | 0 (0.0%) | 2 (8.3%) | 0.493 |
| SOFA score | 6.3 (0.7) | 6.0 (0.5) | 0.935 |
| APACHE II score | 17.3 (1.6) | 18.9 (1.5) | 0.422 |
| Albumin, g/L | 32.1 (5.1) | 31.0 (3.7) | 0.442 |
| Globulin, g/L | 21.8 (3.9) | 23.1 (5.9) | 0.489 |
| Absolute lymphocyte count, $10^9$ /L | 0.9 (0.7) | 0.7 (0.3) | 0.909 |
| White blood cells, $10^9$ /L | 11.0 (5.7) | 12.7 (6.3) | 0.210 |
| CRP, mg/L | 94.7 (101.2) | 108.3 (84.0) | 0.424 |

SOFA score and APACHE II score are mean (SEM), other data are mean (SD), n (%). Mean (SEM/SD) is compared using Mann-Whitney test, and n (%) is compared using Fisher's exact test. * indicated p-value <0.05.

even with accurate pathogenic information. There were a few cases (e.g. B07, B25, and B35 patients) showing no signs of clearing the bacterial infection. (2) The 10 designed pathogens were originated from sepsis, which might not completely overlap with pathogens of severe pneumonia, though pneumonia is one of the most common causes of sepsis. The panel pathogens could be optimized flexibly

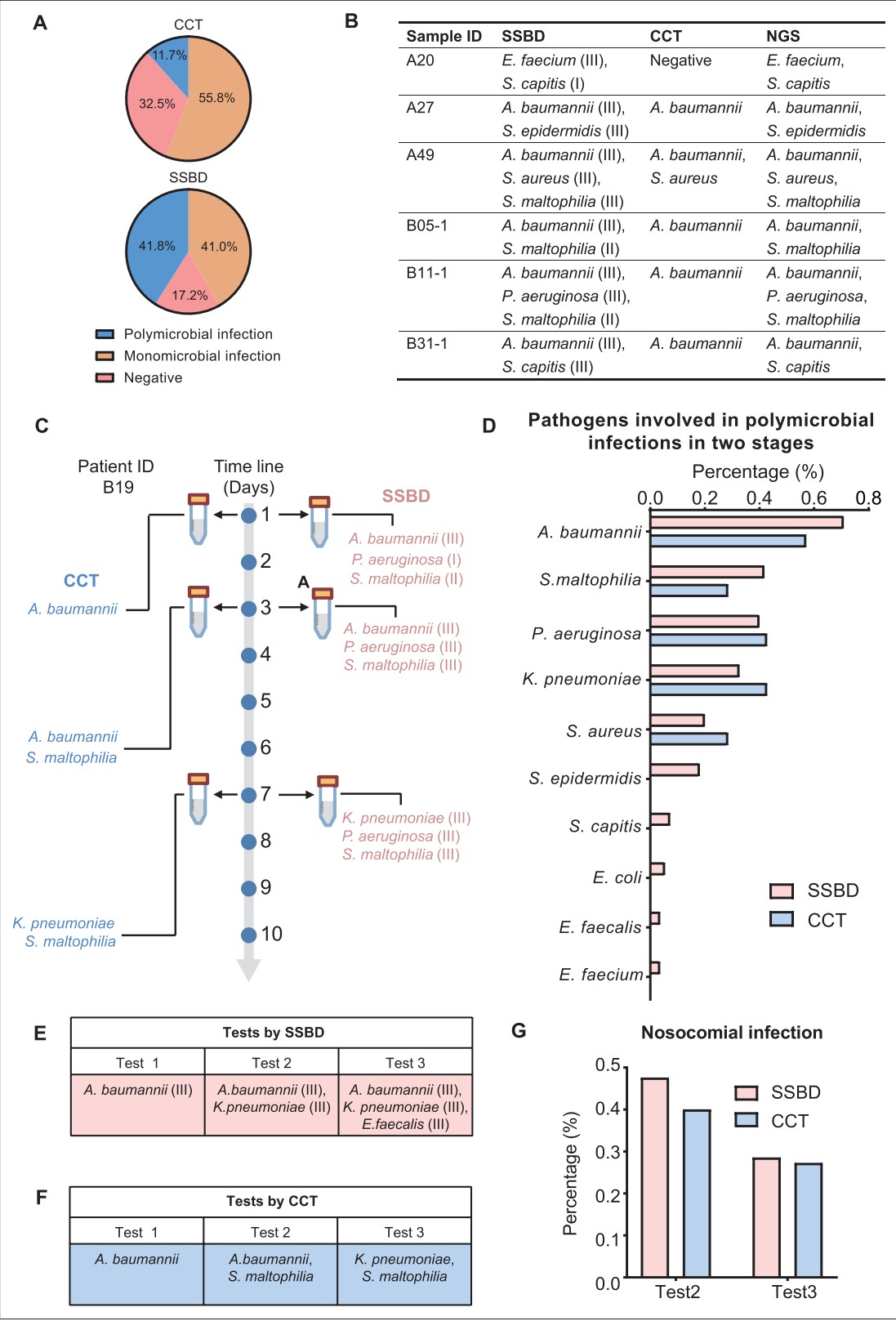

**Figure 5.** Statistical analysis of polymicrobial infection and nosocomial infection in the two validation stages. (**A**) Statistics of pathogenic infection status of BALF samples in the two validation stages. (**B**) Verification from NGS results for 6 samples identified as polymicrobial infection by SSBD but not CCT or missed pathogens by CCT. (**C**) Case study of polymicrobial infection detected by SSBD and CCT. (**D**) Statistics of pathogens involved in polymicrobial infections in the two stages. (**E**) Case study of nosocomial infection identified by SSBD. (**F**) Case study of nosocomial infection identified by CCT. (**G**) Percentage of nosocomial infection identified by SSBD and CCT.

for meeting diverse clinical needs in the ICU. (3) Other potential pathogenic microbes, such as viruses and fungus, might affect the clinical outcomes considering the complexity of ICU patients.

Previous studies showed that polymicrobial pneumonia is related to an increased risk of inappropriate antimicrobial treatment (*Karner et al., 2020*). In both phases, a total of 55 samples were identified as polymicrobial infections by SSBD, while only 14 samples were identified as polymicrobial infections by CCT, which suggested that SSBD could provide more precise pathogenic bacteria information than CCT, especially for those patients with polymicrobial infections. On the other hand, nosocomial infections contribute to a considerable proportion of deaths in ICU patients with severe pneumonia (*Zaragoza et al., 2020*). Although SSBD identified similar ratio of nosocomial infection events with CCT (*Figure 5G*), SSBD provided more timely information for clinical control and response, which might improve the clinical medication decision in ICU.

As anticipated, SSBD performed well with high sensitivity and specificity in rapid pathogens identification, and it possessed shorter turnover time, which was associated with more rapid administration of appropriate antimicrobial therapy in the experiment cases. SSBD also has enormous potential in expanding pathogens from different diseases with much more pathogen genomes included. We believe that SSBD is an accurate tool with great potential but need to be applied in more clinical research.

## Additional information

### Funding

| Funder | Grant reference number | Author |
| --- | --- | --- |
| National Natural Science Foundation of China | The National Key Scientific Instrument and Equipment Development Project. | Wenkui Yu |
| National Natural Science Foundation of China | 81927808 | Wenkui Yu |

The funders had no role in study design, data collection and interpretation, or the decision to submit the work for publication.

### Author contributions

Yan Wang, Resources, Project administration; Xiaohui Liang, Validation, Writing – original draft; Yuqian Jiang, Tianqiang Song, Software; Danjiang Dong, Ming Chen, Han Liu, Haibin Dai, Wanqing Zhou, Supervision; Cong Zhang, Fengchan Xi, Methodology; Yong You, Investigation; Min Ge, Jian-Qun Chen, Writing – review and editing; Qiang Wang, Data curation, Methodology, Writing – original draft; Qihan Chen, Supervision, Writing – review and editing; Wenkui Yu, Methodology, Writing – review and editing

### Author ORCIDs

Xiaohui Liang  http://orcid.org/0000-0001-8065-8168
Qiang Wang  http://orcid.org/0000-0003-2907-9851
Qihan Chen  http://orcid.org/0000-0002-0062-8434
Wenkui Yu  http://orcid.org/0000-0003-4218-0321

### Ethics

Clinical trial registration NCT04178382.
We acquired the ethics approval (2019-197-01) from the ethics committee of Nanjing Drum Tower Hospital Affiliated to Nanjing University Medical School in July 2019, registered and posted the complete research protocol, informed consent, subject materials, case report form, researcher manual, the introduction of main researchers and other information in Chinese. Later on, this study was registered in English at https://clinicaltrials.gov/ (NCT04178382) in November 2019.

### Decision letter and Author response

Decision letter https://doi.org/10.7554/eLife.79014.sa1
Author response https://doi.org/10.7554/eLife.79014.sa2

## Additional files

### Supplementary files
• Appendix 1—figure 2—source data 1. Epidemic data of pathogens in the ICU of Nanjing Drum Tower Hospital in 2017.
• Appendix 1—figure 3—source data 1. The test of SSBD with or without DNA purification.
• Appendix 1—figure 3—source data 2. The test of SSBD with different incubation times.
• Appendix 1—figure 3—source data 3. The validation of SSBD with clinical strains.
• Appendix 1—figure 4—source data 1. Classification of antibiotic coverage for each test in two groups.
• MDAR checklist

### Data availability
All data generated or analysed during this study are included in the manuscript and supporting file; Source Data files have been provided for Figures 3-5, Appendix figures 2-5, and Appendix tables 3-9.

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

## Appendix 1

### Principles of screening species-specific DNA-tags

Our core principles for screening species-specific DNA fragments were as follows: (1) Multiple isolated strains from the same bacterial species were included to ensure the intra-species conservation of selected DNA fragments; (2) Those similar DNA fragments of intra-species conserved fragments in genomes from other bacteria were excluded to ensure inter-species specificity. We developed an original workflow and optimized the algorithm to more efficiently achieve our purposes compared with conventional pairwise alignment (*Appendix 1—figure 1A*). Firstly, conserved DNA regions were obtained by aligning the genomes of two strains within a given species, downsizing the genome to the regional scale. Those conserved DNA regions were performed alignments with the genomes of other strains from the same species, only shared DNA regions in all the strains are retained, downsizing conserved genomic regions to intra-species conserved DNA fragments (*Appendix 1—figure 1B*). Secondly, to achieve inter-species specificity of DNA tags, intra-species conserved DNA fragments were performed alignments with the genomes of other bacterial species. Those similar fragments of intra-species conserved DNA fragments in genomes from other bacteria species were excluded. After two-step screening, we finally obtained species-specific DNA tags (*Appendix 1—figure 1C*). The software for screening species-specific DNA fragments could be obtained from the following URL: https://github.com/wang-q/App-Egaz; (*Wang, 2022* copy archived at swh:1:rev:58b19aa6a5d540030ff43796c141e810cced07ce).

### Sample size and randomized double-blind trial

Based on the local epidemiology, the mortality rate of the early stage of sepsis in the adequate anti-infective treatment group was 48%, and was 65% in the inadequate anti-infective treatment group. The sample size was estimated through the formula: $\alpha$=0.05, 1- $\beta$=0.80, Pt: 0.65, Pc: 0.42, Nt/Nc = 1, Power = 0.802, H0: Pt - Pc = 0; H1: Pt - Pc ≠ 0. Nt = 73, Nc = 73. The sample size for the clinical study was finally determined to be 73 patients for each group. The randomized process was conducted by SAS 9.4 software to determine whether patients were selected for the experiment group or the control group. The random numbers were generated using the PLAN process. Then all numbers are concealed in random envelopes. Patients were assigned by opening the random number in random envelopes. For CCT and SSBD detection, the two processes were carried out independently in the hospital microbiology laboratory and research lab, which were blinded to each other. The detection results were finally unblinded by the clinician of ICU with non-testing procedures.

### Extracting samples

Extracting bronchoalveolar lavage fluid (BALF) for the enrolled patients were under mild anesthesia via tracheal intubation or tracheotomy entering the infected bronchus. 30 mL saline was injected into batches quickly and recollected by using negative pressure lower than 13.3 kPa. In the validation stage I, 5 mL collected samples were cultivated as usual, and another 5 mL samples were sent for SSBD testing. As a liquid biopsy technique, SSBD was not involved any additional operations that could harm the patients' health compared to CCT.

### Reaction process

Briefly, DNA of BALF samples was extracted using Quick–DNA/RNA Pathogen Miniprep Kit (Zymo Research) according to the manual and diluted to 100 ng/μL. The DNA samples were then amplified using designed specific primers for 10 bacteria. Then, PCR products, Cas12a-crRNA complexes and the reporter DNA probe were added to the reaction system. Finally, the fluorescence signals were detected after incubation for 30 mins at 37 °C. Some samples were sent for next-generation sequencing (IngeniGen XunMinKang Biotechnology Inc Hangzhou, Zhejiang, China).

### Clinical operation process

In the validation stage II, BALFs were collected from each patient on the first day, day 3–5, and day 7+unless the airway was removed. All samples were sent for microbial cultivation, and simultaneously the samples taken from the experiment group were tested by SSBD. Once BALF results were available, at least two clinical experts (antimicrobial stewardship) discussed and decided on antibiotics adjustment according to the results and other clinical data. Additionally, the experts would assess each patient's receiving antibiotics coverage rate at different times and other clinical outcomes retrospectively.

Treatment and outcomes data, such as evaluation of therapeutic effectiveness at day 3, 7, 10 and 14, time of mechanical ventilation and vasopressor support from enrollment to day 28, occurrence of antibiotic-associated diarrhea, and mortality on day 28 were also recorded and analyzed. Evaluation of therapeutic effectiveness was conducted by two senior clinicians according to these parameters.: (1) whether fever and purulent secretion were improved or not; (2) whether leukocytosis or leukopenia got better or not; (3) whether radiological pulmonary infiltrate absorbed partly or not; (4) whether oxygenation index was improved or not; (5) whether hemodynamic instability was rectified gradually or not.

## SSBD diagnostic report

For every single experiment, we test the fluorescence value of clinical separated positive bacteria strains as positive control (PC) and DEPC-H2O as negative control (NC). For each experiment, we test the fluorescence value of BALF (shown as F) with a microplate reader. When F/NC >2, it is the signal that bacteria detected by our method. We use I (interval) as our point of distinction. I = (PC - 2NC) / 3. When 2NC <F < 2 NC+I, the bacteria strength level is defined as level I. When 2NC + I < F<2 NC+2 I, the bacteria strength level is defined as level II. When $F$>2 NC+2 I, the bacteria strength level is defined as level III. Different levels of bacterial strength can roughly represent different bacterial copies according to our method lowest detection rate (Level I: $10^{-15}$ M–$10^{-14}$ M, Level II: $10^{-14}$ M–$10^{-13}$ M, Level III: over $10^{-13}$ M, *Figure 3B*). We separated different testing values as different levels, suggesting that our method could test pathogen strength to some extent and give pathogens strength for drug usage. If the sample was discovered to be affected by any unexpected condition (e.g. contamination during the test, etc.), the detection results of the sample would be excluded from comparative analysis of diagnostic accuracy. If any missing data was critical for result judgment, such result would be excluded from subsequent analysis and statistics.

## Evaluation of antibiotics coverage

For three individually designated BALF tests, the rate of antibiotics coverage was calculated in all groups. We calculated coverage rate from two aspects. For each test, we calculated rate of covered samples among all samples tested. For each patient, we calculated their coverage rate by counting covered test numbers within all tests that had been taken. Evaluation of antibiotics coverage was made by two experts retrospectively, according to microbial, antimicrobial susceptibility tests (AST) and clinical treatment effect. The details were as follow:

1. Our test result is negative (from the experiment group), or microbial cultivation is negative (from the control group). If it is deemed effective on the clinical signs (1), it is judged as antibiotics covering. If it is deemed clinically invalid, it is judged as antibiotics uncovering.
2. Our test result is positive (from the experiment group), or microbial cultivation is positive (from the control group). If AST (from experiment group or control group) is shown sensitive to the antibiotics, no matter the clinical signs are effective or not, it is judged as antibiotics covering. If AST is shown resistant to using antibiotics, whether antibiotics covering or not is judged by clinical effectiveness. It is identified as covered when clinically effective and uncovered when clinically invasive.

When BALF is not collected for a test or microbial cultivation from the day on, whether with covering antibiotics or not on the day is judged on clinical effectiveness.

**A**

Conventional pairwise alignment

Optimized alignment method

$$N_{SA} = \frac{n^2}{2} - \frac{n}{2}$$

$$N_{SA} = n - 1$$

**B**

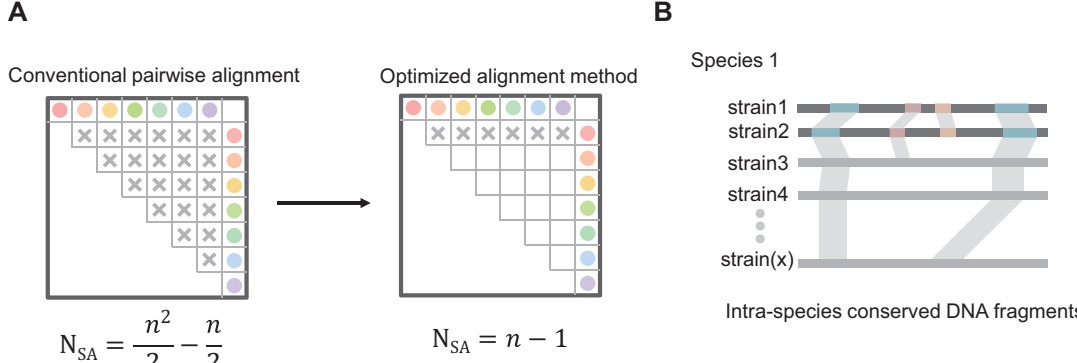

Species 1

strain1
strain2
strain3
strain4
⋮
strain(x)

Intra-species conserved DNA fragments

**C**

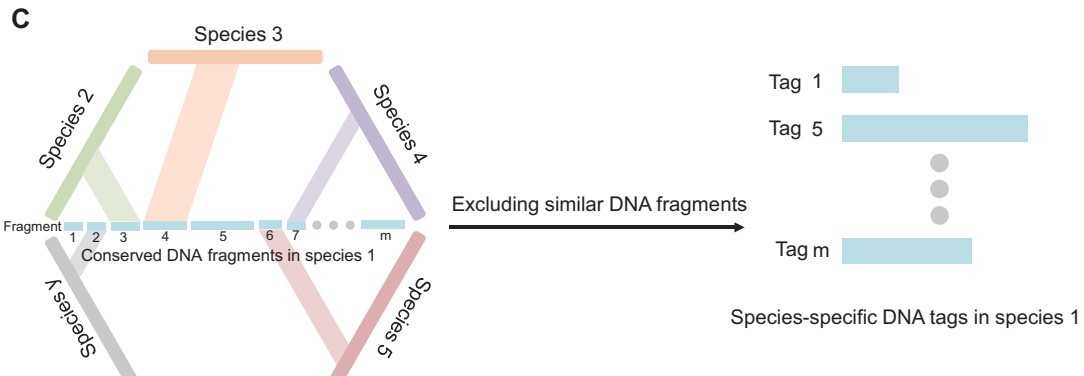

Species 3

Species 2

Species 4

Fragment
1 2 3 4 5 6 7 ... m
Conserved DNA fragments in species 1

Species y

Species 5

Excluding similar DNA fragments

Tag 1
Tag 5
⋮
Tag m

Species-specific DNA tags in species 1

**Appendix 1—figure 1.** Diagram of core principles for screening species-specific DNA-tags. (**A**) Optimizing the algorithm of sequence alignment. Abbreviations: SA, sequence alignment; N, number of double sequence alignment; n, number of sequences. (**B**) Schematic map of screening intra-species conserved DNA fragments. (**C**) Schematic map of screening species-specific DNA tags.

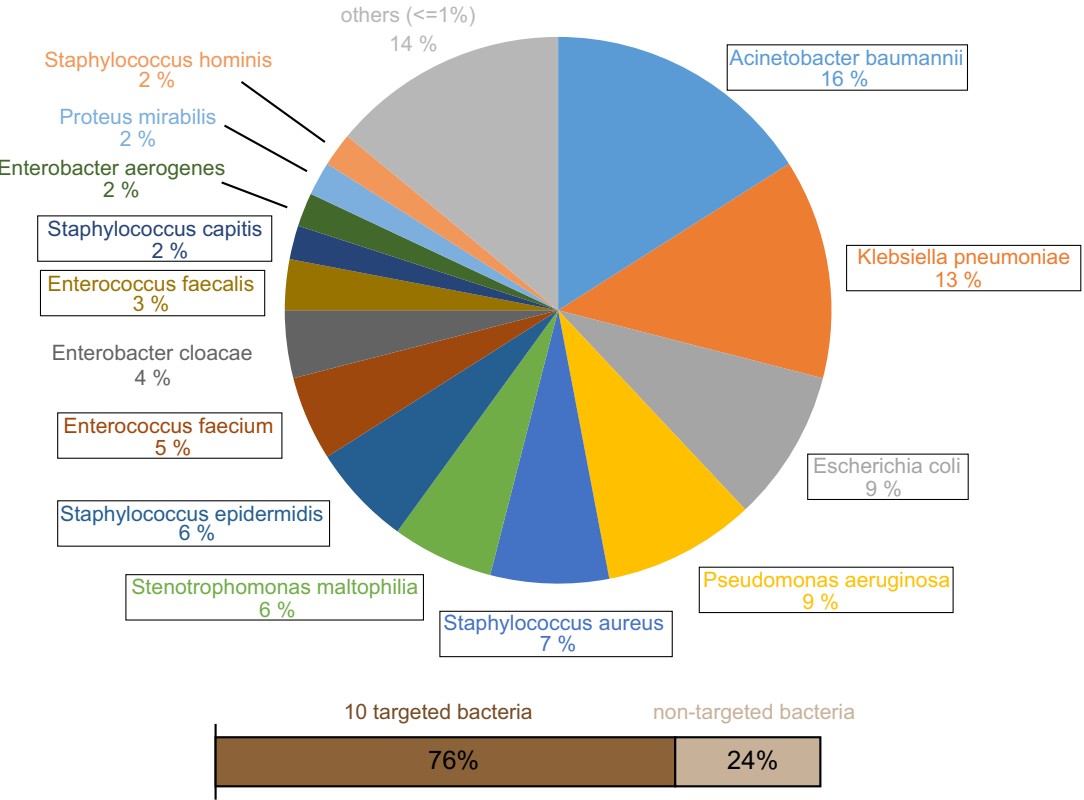

## Pathogenic bacteria in ICU of Nanjing Drum Tower Hospital (2017)

**Appendix 1—figure 2.** Epidemic data of pathogens in the Nanjing Drum Tower Hospital ICU in 2017. 10 targeted bacteria were indicated with the box.

The online version of this article includes the following source data for appendix 1—figure 2:

**Appendix 1—figure 2—source data 1.** Epidemic data of pathogens in the ICU of Nanjing Drum Tower Hospital in 2017.

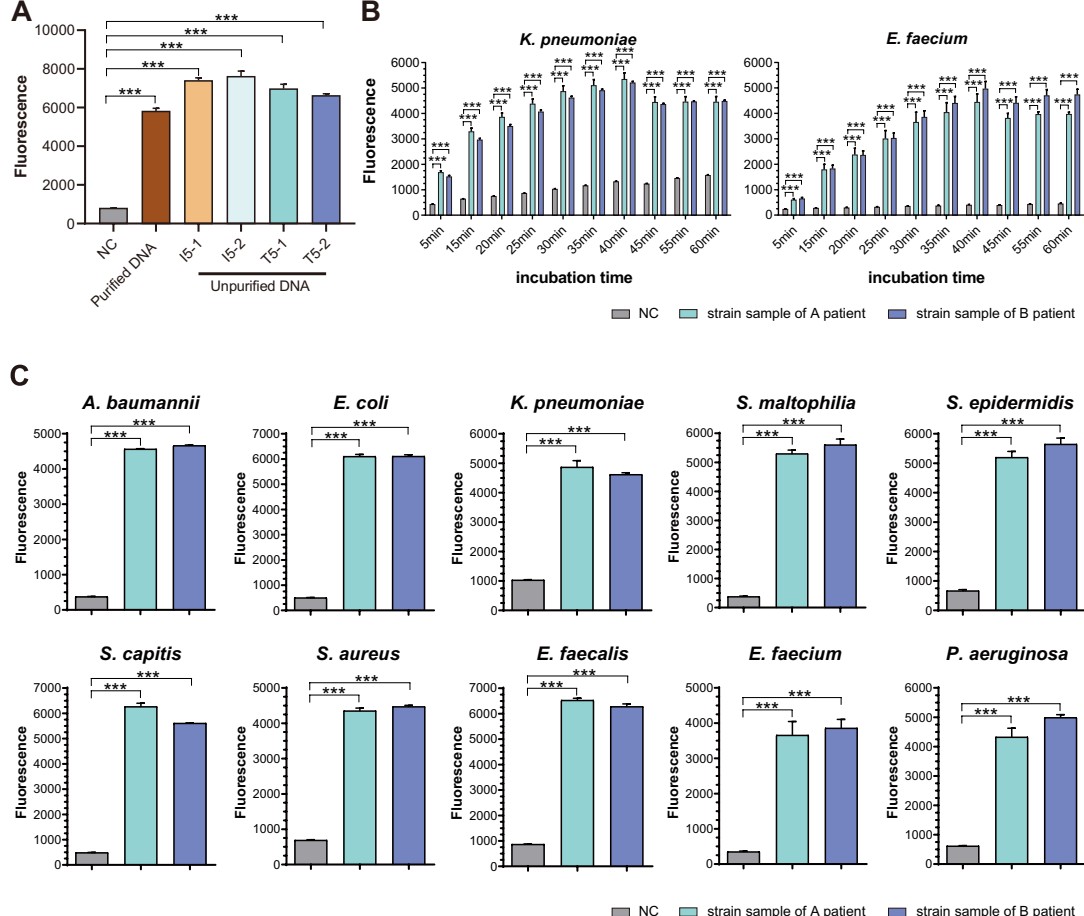

**Appendix 1—figure 3.** SSBD development and effectiveness validation. (**A**) SSBD results of purified and unpurified DNA. NC, namely the fluorescence values of PCR products of using DEPC-H2O as input. Each group had three repeats. Error bars indicated mean ± SEM of fluorescence value. *** indicated p-value <0.001 of unpaired t-test. (**B**) SSBD results of reaction time gradient with Cas12a. Fluorescence values of *K. pneumoniae* and *E. faecium* by Cas12a through different incubation times after PCR. Gray represented NC, namely the fluorescence values of PCR products of using DEPC-H2O as input. Green and blue represented the fluorescence values of bacteria strains from different patients. Each group had three repeats. Error bars indicated mean ± SEM of fluorescence value. ** indicated p-value <0.01 and *** indicated p-value <0.001 of unpaired t-test. (**C**) SSBD results of 10 pathogenic bacteria with Cas12a. Gray represented NC, namely the fluorescence values of PCR products of using DEPC-H2O as input. Green and blue represented the fluorescence values of bacteria strains from different patients. Each group had three repeats. Error bars indicated mean ± SEM of fluorescence value. *** indicated p-value <0.001 of unpaired t-test.

The online version of this article includes the following source data for appendix 1—figure 3:

**Appendix 1—figure 3—source data 1.** The test of SSBD with or without DNA purification.

**Appendix 1—figure 3—source data 2.** The test of SSBD with different incubation times.

**Appendix 1—figure 3—source data 3.** The validation of SSBD with clinical strains.

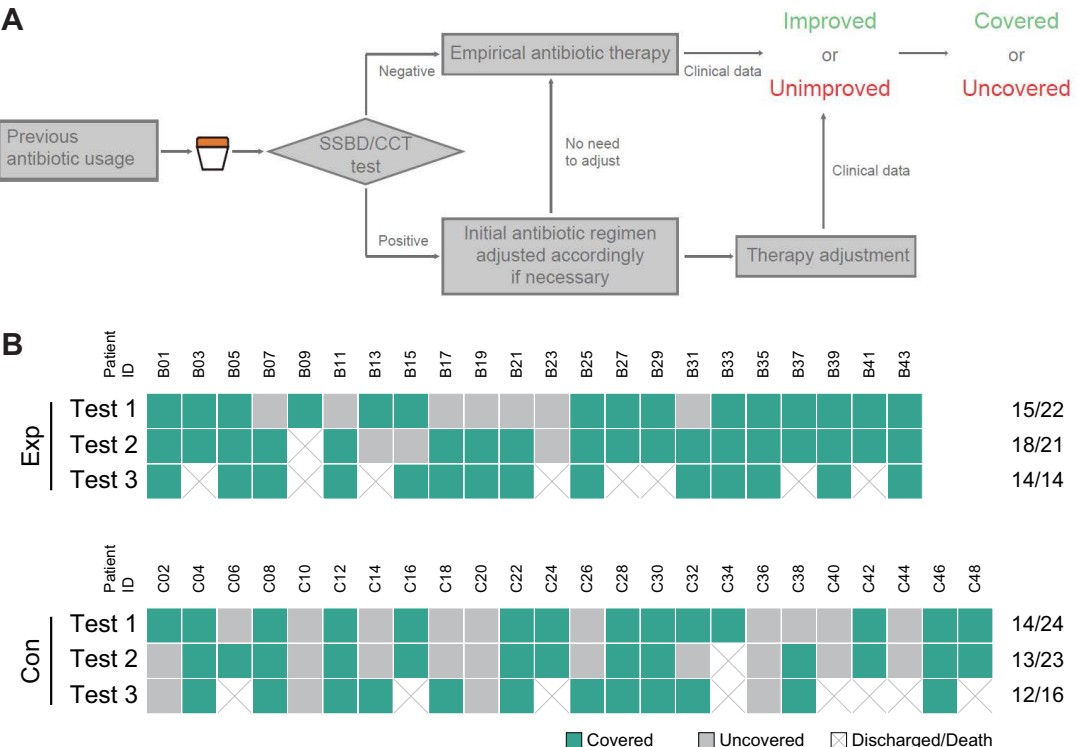

**Appendix 1—figure 4.** Judgment process and results of antibiotics coverage. (**A**) Judgment process of antibiotics coverage. (**B**) The raw results of antibiotics coverage in two groups. Exp meant the experimental group, and Con meant the control group.

The online version of this article includes the following source data for appendix 1—figure 4:

**Appendix 1—figure 4—source data 1.** Classification of antibiotic coverage for each test in two groups.

**A    Validation stage I**

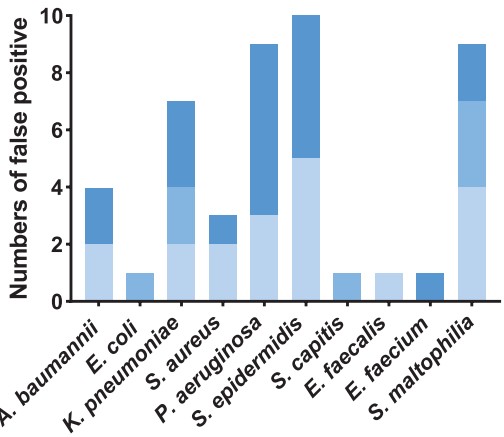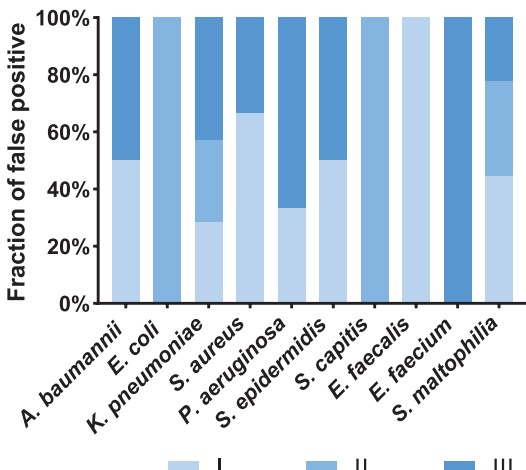

**B    Validation stage II**

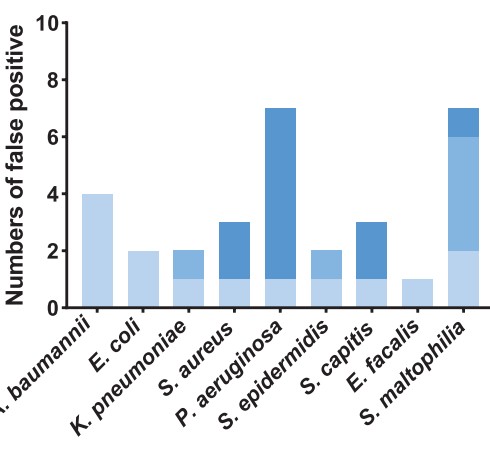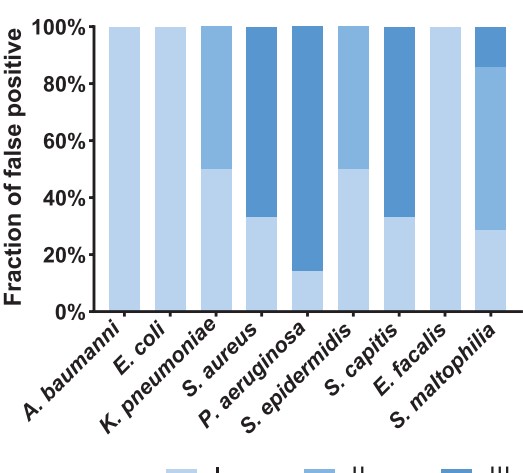

**Appendix 1—figure 5.** Analysis of false-positive samples. Numbers and fractions of different strength levels among all false-positive samples of each bacteria species in the validation stage I (**A**) and II (**B**). Strength could be seen roughly as bacterial amounts (level I-level III, the definition could be seen in the Appendix 1). False-positive situations meant pathogenic bacteria detected by SSBD but not by CCT in a given BALF sample.

**Appendix 1—table 1.** Primers used in experiments.

| Name | Sequence (5' ->3') | Target | Product length (bp) |
| --- | --- | --- | --- |
| pGL3-amplify-F | GAAGATGGAACCGCTGGAGA | pGL3 | 597 |
| pGL3-amplify-R | GCAGGCAGTTCTATGAGGCA | | |
| Aba-amplify-F | CACAGCGTTTAACCCATGCC | *A. baumannii* | 564 |
| Aba-amplify-R | TATCGCCACCTGCACAGAAG | | |
| Eco-amplify-F | GTTCCTGACTATCTGGCGGG | *E. coli* | 371 |
| Eco-amplify-R | GCTTCCTGACTCCAGACACC | | |

*Appendix 1—table 1 Continued on next page*

*Appendix 1—table 1 Continued*

| Name | Sequence (5' ->3') | Target | Product length (bp) |
|---|---|---|---|
| Kpn-amplify-F | CATGGGCATATCGACGGTCA | *K. pneumoniae* | 740 |
| Kpn-amplify-R | CCTGCAACATAGGCCAGTGA | | |
| Sau-amplify-F | AGGTGCAGTAGACGCATAGC | *S. aureus* | 563 |
| Sau-amplify-R | CATTCGCTGCGCCAATACAA | | |
| Pae-amplify-F | TCTCTCTATCACGCCGGTCA | *P. aeruginosa* | 467 |
| Pae-amplify-R | TCGCATCGAGGTATTCCAGC | | |
| Sep-amplify-F | CACGCATGGCACTAGGTACA | *S. epidermidis* | 383 |
| Sep-amplify-R | CGAAAAAGAGTTGTCCTTGTTGA | | |
| Sca-amplify-F | GGTTCAGTCATCCCCACGTT | *S. capitis* | 591 |
| Sca-amplify-R | CAGCTGCGACAACTGCTTAC | | |
| Efa-amplify-F | CGGCAAGTTTGGAAGCAGAC | *E. faecalis* | 627 |
| Efa-amplify-R | CAGCGCCTAGTCCTTGTGAT | | |
| Efm-amplify-F | ATCGGAAATCGGTGTGGCTT | *E. faecium* | 507 |
| Efm-amplify-R | TCAAATGCATCCCTGTGCCT | | |
| Sma-amplify-F | CGCCTCCCGTTTACAGATTA | *S. maltophilia* | 356 |
| Sma-amplify-R | TCGGCTCCACCACATACAC | | |

## Appendix 1—table 2. Oligonucleotide templates for synthesis of crRNAs.

| Name | Sequence (5'->3') | Target |
|---|---|---|
| T7-Forward | TAATACGACTCACTATAGGT | |
| LbcrRNA-Reverse-PGL3 | ATGTGACGAACGTGTACATCGACTATCTACACTTAGTAGAAATTACCTATAGTGAGTCGTATTA | PGL3 |
| LbcrRNA-Reverse-Aba | TTCAAGTAATTCTTCTTTACATCTACACTTAGTAGAAATTACCTATAGTGAGTCGTATTA | *A. baumannii* |
| LbcrRNA-Reverse-Eco | CTTGCCATCATAGCGACCGTATCTACACTTAGTAGAAATTACCTATAGTGAGTCGTATTA | *E. coli* |
| LbcrRNA-Reverse-Kpn | AAGATGGCGATTACCGCAGTATCTACACTTAGTAGAAATTACCTATAGTGAGTCGTATTA | *K. pneumoniae* |
| LbcrRNA-Reverse-Sau | CCTCAGCAAGTTCACGTTGTATCTACACTTAGTAGAAATTACCTATAGTGAGTCGTATTA | *S. aureus* |
| LbcrRNA-Reverse-Pae | CTCCTCATGTGTGTTTACAAATCTACACTTAGTAGAAATTACCTATAGTGAGTCGTATTA | *P. aeruginosa* |
| LbcrRNA-Reverse-Sep | TCTCTAATTGATGAATATTAATCTACACTTAGTAGAAATTACCTATAGTGAGTCGTATTA | *S. epidermidis* |
| LbcrRNA-Reverse-Sca | TATTGATTAATAAGGTGATTATCTACACTTAGTAGAAATTACCTATAGTGAGTCGTATTA | *S. capitis* |
| LbcrRNA-Reverse-Efa | TCAGCTGTGTTATTTGGTGCATCTACACTTAGTAGAAATTACCTATAGTGAGTCGTATTA | *E. faecalis* |
| LbcrRNA-Reverse-Efm | TGTATATAAGTTCAGGTAGTATCTACACTTAGTAGAAATTACCTATAGTGAGTCGTATTA | *E. faecium* |
| LbcrRNA-Reverse-Sma | CCTGGTCGCAGGTGTCATGCATCTACACTTAGTAGAAATTACCTATAGTGAGTCGTATTA | *S. maltophilia* |

## Appendix 1—table 3. SSBD and CCT results of BALF samples in the validation stage I.

| Sample ID | SSBD Results | CCT Results | NGS Results |
|---|---|---|---|
| A01 | *S. aureus* (I) | N | *S. aureus* |
| A02 | *A. baumannii* (III) | *A. baumannii* | |

*Appendix 1—table 3 Continued on next page*

*Appendix 1—table 3 Continued*

| Sample ID | SSBD Results | CCT Results | NGS Results |
|---|---|---|---|
| A03 | *A. baumannii* (III)<br>*S. aureus* (I) | *A. baumannii*<br>*S. aureus* | *A. baumannii*<br>*S. aureus* |
| A04 | *A. baumannii* (III)<br>*K. pneumoniae* (II)<br>*S. maltophilia* (II) | N | |
| A05 | *A. baumannii* (III)<br>*K. pneumoniae* (III)<br>*P. aeruginosa* (III) | *K. pneumoniae*<br>*P. aeruginosa* | |
| A06 | *A. baumannii* (III) | *A. baumannii* | |
| A07 | *S. maltophilia* (II) | *S. maltophilia* | |
| A08 | *A. baumannii* (III) | *A. baumannii* | |
| A09 | *A. baumannii* (II)<br>*S. aureus* (I) | *A. baumannii* | |
| A10 | *A. baumannii* (III)<br>*K. pneumoniae* (III)<br>*P. aeruginosa* (I) | *A. baumannii*<br>*K. pneumoniae* | |
| A11 | *A. baumannii* (III)<br>*K. pneumoniae* (I) | *A. baumannii* | |
| A12 | N | N | |
| A13 | *A. baumannii* (II) | *A. baumannii* | |
| A14 | N | N | |
| A15 | *K. pneumoniae* (III) | *K. pneumoniae* | |
| A16 | *K. pneumoniae* (III)<br>*P. aeruginosa* (III) | *K. pneumoniae* | |
| A17 | N | N | |
| A18 | *K. pneumoniae* (I)<br>*P. aeruginosa* (III) | *K. pneumoniae*<br>*P. aeruginosa* | |
| A19 | *P. aeruginosa* (I) | *P. aeruginosa* | |
| A20 | *E. faecium* (III)<br>*S. capitis* (II) | N | *E. faecium*<br>*S. capitis* |
| A21 | *A. baumannii* (III)<br>*K. pneumoniae* (III)<br>*S. aureus* (I) | *A. baumannii*<br>*S. aureus* | |
| A22 | *A. baumannii* (III)<br>*S. epidermidis* (III) | *A. baumannii* | |
| A23 | N | N | |
| A24 | *A. baumannii* (III) | *A. baumannii* | |
| A25 | N | N | N |
| A26 | *P. aeruginosa* (III) | *P. aeruginosa* | |
| A27 | *A. baumannii* (III)<br>*S. epidermidis* (III) | *A. baumannii* | *A. baumannii*<br>*S. epidermidis* |
| A28 | *A. baumannii* (II)<br>*P. aeruginosa* (III) | *A. baumannii* | |
| A29 | *S. epidermidis* (I)<br>*S. maltophilia* (I) | N | |

*Appendix 1—table 3 Continued on next page*

*Appendix 1—table 3 Continued*

| Sample ID | SSBD Results | CCT Results | NGS Results |
|---|---|---|---|
| A30 | A. baumannii (III)<br>S. aureus (III)<br>S. epidermidis (I) | A. baumannii | |
| A31 | K. pneumoniae (I)<br>P. aeruginosa (I)<br>S. epidermidis (I) | N | K. pneumoniae |
| A32 | P. aeruginosa (I)<br>S. epidermidis (I) | N | |
| A33 | S. aureus (I)<br>S. maltophilia (II) | S. aureus | |
| A34 | N | N | A. baumannii |
| A35 | A. baumannii (III) | A. baumannii | |
| A36 | A. baumannii (III)<br>S. maltophilia (I) | A. baumannii | |
| A37 | A. baumannii (III)<br>S. epidermidis (III) | A. baumannii | |
| A38 | A. baumannii (III) | A. baumannii | |
| A39 | A. baumannii (III) | A. baumannii | |
| A40 | N | N | |
| A41 | N | N | |
| A42 | A. baumannii (I) | N | |
| A43 | A. baumannii (III) | A. baumannii | |
| A44 | A. baumannii (III) | A. baumannii | A. baumannii |
| A45 | A. baumannii (III)<br>K. pneumoniae (II) | A. baumannii | |
| A46 | A. baumannii (III) | A. baumannii | |
| A47 | A. baumannii (I) | A. baumannii | |
| A48 | A. baumannii (III) | A. baumannii | |
| A49 | A. baumannii (III)<br>S. aureus (III)<br>S. maltophilia (III) | A. baumannii<br>S. aureus | A. baumannii<br>S. aureus<br>S. maltophilia |
| A50 | S. epidermidis (III) | N | |
| A51 | N | N | |
| A52 | N | N | |
| A53 | K. pneumoniae (III) | K. pneumoniae | |
| A54 | N | N | |
| A55 | A. baumannii (III)<br>E. coli (II)<br>K. pneumoniae (III)<br>S. maltophilia (III) | A. baumannii | |
| A56 | S. epidermidis (III)<br>E. faecalis (I) | N | |
| A57 | N | N | |
| A58 | P. aeruginosa (III) | N | |

*Appendix 1—table 3 Continued on next page*

*Appendix 1—table 3 Continued*

| Sample ID | SSBD Results | CCT Results | NGS Results |
|---|---|---|---|
| A59 | *S. maltophilia* (I) | *S. maltophilia* | *S. aureus*<br>*S. maltophilia* |
| A60 | *A. baumannii* (I)<br>*P. aeruginosa* (III) | *A. baumannii*<br>*P. aeruginosa* | |
| A61 | *A. baumannii* (II) | *A. baumannii* | |
| A62 | *K. pneumoniae* (II)<br>*S. maltophilia* (III) | *K. pneumoniae*<br>*S. maltophilia* | |
| A63 | *A. baumannii* (III) | *A. baumannii* | |
| A64 | *A. baumannii* (III) | *A. baumannii* | |
| A65 | *A. baumannii* (III)<br>*P. aeruginosa* (I)<br>*S. maltophilia* (II) | *A. baumannii*<br>*P. aeruginosa* | |
| A66 | *K. pneumoniae* (II) | *K. pneumoniae* | |
| A67 | *S. aureus* (III) | *S. aureus* | |
| A68 | *A. baumannii* (III)<br>*P. aeruginosa* (III) | *A. baumannii* | |
| A69 | *A. baumannii* (I)<br>*P. aeruginosa* (III) | *A. baumannii* | *P. aeruginosa* |
| A70 | *A. baumannii* (I) | *A. baumannii* | |
| A71 | N | N | |
| A72 | *A. baumannii* (III)<br>*S. maltophilia* (I) | *A. baumannii* | |
| A73 | *A. baumannii* (III)<br>*K. pneumoniae* (III)<br>*P. aeruginosa* (III) | *A. baumannii*<br>*P. aeruginosa* | |
| A74 | *A. baumannii* (III) | *A. baumannii* | |
| A75 | *A. baumannii* (III)<br>*P. aeruginosa* (III) | *A. baumannii* | |
| A76 | *A. baumannii* (I)<br>*P. aeruginosa* (III)<br>*S. maltophilia* (I) | *P. aeruginosa* | |
| A77 | *S. epidermidis* (I) | N | |

**Appendix 1—table 4.** Comparative analysis of test results by SSBD, CCT and NGS in the validation stage I.

| Sample ID | SSBD Results | CCT Results | NGS Results |
|---|---|---|---|
| A01 | *S. aureus* (I) | N | *S. aureus* |
| A03 | *A. baumannii* (III)<br>*S. aureus* (I) | *A. baumannii*<br>*S. aureus* | *A. baumannii*<br>*S. aureus* |
| A20 | *E. faecium* (III)<br>*S. capitis* (II) | N | *E. faecium*<br>*S. capitis* |
| A25 | N | N | N |
| A27 | *A. baumannii* (III)<br>*S. epidermidis* (III) | *A. baumannii* | *A. baumannii*<br>*S. epidermidis* |
| A31 | *K. pneumoniae* (I)<br>*P. aeruginosa* (I)<br>*S. epidermidis* (I) | N | *K. pneumoniae* |

*Appendix 1—table 4 Continued on next page*

*Appendix 1—table 4 Continued*

| Sample ID | SSBD Results | CCT Results | NGS Results |
|---|---|---|---|
| A34 | N | N | A. baumannii |
| A44 | A. baumannii (III) | A. baumannii | A. baumannii |
| A49 | A. baumannii (III)<br>S. aureus (III)<br>S. maltophilia (III) | A. baumannii<br>S. aureus | A. baumannii<br>S. aureus<br>S. maltophilia |
| A59 | S. maltophilia (I) | S. maltophilia | S. maltophilia<br>S. aureus |
| A69 | A. baumannii (I)<br>P. aeruginosa (III) | A. baumannii | P. aeruginosa |

**Appendix 1—table 5.** SSBD and CCT results of BALF samples from experiment group (n=22) and control group (n=24) during the validation stage II.

| Patient ID | Sample ID | Test No. | SSBD Results | CCT Results | NGS Results |
|---|---|---|---|---|---|
| **Exp.** | | | | | |
| B01 | B01-1 | Test 1 | N | N | |
| | B01-2 | Test 2 | N | N | N |
| | B01-3 | Test 3 | A. baumannii (III) | A. baumannii | |
| B03 | B03-1 | Test 1 | N | N | N |
| | B03-2 | Test 2 | A. baumannii (III) | A. baumannii | |
| B05 | B05-1 | Test 1 | A. baumannii (III)<br>S. maltophilia (II) | A. baumannii | A. baumannii<br>S. maltophilia |
| | B05-2 | Test 2 | A. baumannii (III)<br>S. maltophilia (I) | A. baumannii | |
| | B05-3 | Test 3 | A. baumannii (III)<br>S. maltophilia (I) | | |
| B07 | B07-1 | Test 1 | P. aeruginosa (III) | P. aeruginosa | P. aeruginosa |
| | B07-2 | Test 2 | P. aeruginosa (III) | P. aeruginosa | |
| | B07-3 | Test 3 | P. aeruginosa (III) | P. aeruginosa | |
| B09 | B09-1 | Test 1 | A. baumannii (III)<br>S. aureus (III)<br>S. capitis (III)<br>S. maltophilia (III) | A. baumannii | |
| B11 | B11-1 | Test 1 | A. baumannii (III)<br>P. aeruginosa (III)<br>S. maltophilia (II) | A. baumannii | A. baumannii<br>P. aeruginosa<br>S. maltophilia |
| | B11-2 | Test 2 | A. baumannii (III)<br>S. aureus (I)<br>S. maltophilia (III) | | |
| | B11-3 | Test 3 | P. aeruginosa (III) | N | |
| B13 | B13-1 | Test 1 | S. epidermidis (I) | N | N |
| | B13-2 | Test 2 | K. pneumoniae (I) | K. pneumoniae | |
| B15 | B15-1 | Test 1 | A. baumannii (I)<br>E. coli (I)<br>K. pneumoniae (II) | N | |
| | B15-2 | Test 2 | K. pneumoniae (I)<br>E. faecium (I) | | |
| | B15-3 | Test 3 | E. faecium (III) | | |

*Appendix 1—table 5 Continued on next page*

Appendix 1—table 5 Continued

| Patient ID | Sample ID | Test No. | SSBD Results | CCT Results | NGS Results |
|---|---|---|---|---|---|
| B17 | B17-1 | Test 1 | A. baumannii (III) | | |
| | B17-2 | Test 2 | A. baumannii (III)<br>K. pneumoniae (III) | | |
| | B17-3 | Test 3 | A. baumannii (I)<br>K. pneumoniae (I)<br>E. faecalis (I) | A. baumannii | |
| B19 | B19-1 | Test 1 | A. baumannii (III)<br>P. aeruginosa (I)<br>S. maltophilia (II) | A. baumannii | |
| | B19-2 | Test 2 | A. baumannii (III)<br>P. aeruginosa (III)<br>S. maltophilia (III) | A. baumannii<br>S. maltophilia | |
| | B19-3 | Test 3 | K. pneumoniae (III)<br>P. aeruginosa (III)<br>S. maltophilia (III) | K. pneumoniae<br>S. maltophilia | |
| B21 | B21-1 | Test 1 | P. aeruginosa (III)<br>S. aureus (III)<br>S. maltophilia (III) | S. aureus<br>S. maltophilia | |
| | B21-2 | Test 2 | P. aeruginosa (III)<br>S. aureus (III)<br>S. maltophilia (III) | S. maltophilia | |
| | B21-3 | Test 3 | S. aureus (I)<br>S. maltophilia (II) | N | |
| B23 | B23-1 | Test 1 | S. capitis (I)<br>S. maltophilia (III) | S. maltophilia | |
| | B23-2 | Test 2 | S. maltophilia (III) | S. maltophilia | |
| B25 | B25-1 | Test 1 | A. baumannii (II) | A. baumannii | |
| | B25-2 | Test 2 | A. baumannii (III) | | |
| | B25-3 | Test 3 | A. baumannii (III) | A. baumannii | |
| B27 | B27-1 | Test 1 | S. maltophilia (I) | N | N |
| | B27-2 | Test 2 | A. baumannii (III) | | |
| B29 | B29-1 | Test 1 | K. pneumoniae (III) | K. pneumoniae | K. pneumoniae |
| | B29-2 | Test 2 | K. pneumoniae (III) | | |
| B31 | B31-1 | Test 1 | A. baumannii (III)<br>S. capitis (III) | A. baumannii | A. baumannii<br>S. capitis |
| | B31-2 | Test 2 | A. baumannii (III) | | |
| | B31-3 | Test 3 | A. baumannii (I) | N | |
| B33 | B33-1 | Test 1 | N | N | |
| | B33-2 | Test 2 | A. baumannii (I)<br>S. epidermidis (II) | N | |
| | B33-3 | Test 3 | N | N | |
| B35 | B35-1 | Test 1 | A. baumannii (III) | A. baumannii | A. baumannii |
| | B35-2 | Test 2 | A. baumannii (III) | A. baumannii | |
| | B35-3 | Test 3 | A. baumannii (III) | A. baumannii | |
| B37 | B37-1 | Test 1 | N | N | A. baumannii |
| | B37-2 | Test 2 | P. aeruginosa (I) | | |

Appendix 1—table 5 Continued on next page

*Appendix 1—table 5 Continued*

| Patient ID | Sample ID | Test No. | SSBD Results | CCT Results | NGS Results |
|---|---|---|---|---|---|
| B39 | B39-1 | Test 1 | N | *P. aeruginosa* | |
| | B39-2 | Test 2 | *A. baumannii* (I) *E. coli* (I) *K. pneumoniae* (III) *P. aeruginosa* (III) | *K. pneumoniae* *P. aeruginosa* | |
| | B39-3 | Test 3 | *K. pneumoniae* (I) *P. aeruginosa* (III) | | |
| B41 | B41-1 | Test 1 | N | N | N |
| | B41-2 | Test 2 | N | | |
| B43 | B43-1 | Test 1 | N | N | |
| | B43-2 | Test 2 | *A. baumannii* (I) *S. epidermidis* (I) | | |
| | B43-3 | Test 3 | *A. baumannii* (II) | *A. baumannii* | |
| **Con**. | | | | | |
| C02 | | Test 1 | | *A. baumannii* | |
| | | Test 2 | | *A. baumannii* | |
| | | Test 3 | | *A. baumannii* | |
| C04 | | Test 1 | | N | |
| | | Test 2 | | N | |
| | | Test 3 | | N | |
| C06 | | Test 1 | | N | |
| | | Test 2 | | *A. baumannii* | |
| C08 | | Test 1 | | *E. coli* | |
| | | Test 2 | | *E. coli* | |
| | | Test 3 | | *K. pneumoniae* | |
| C10 | | Test 1 | | N | |
| | | Test 2 | | N | |
| | | Test 3 | | *A. baumannii* | |
| C12 | | Test 1 | | N | |
| | | Test 2 | | *A. baumannii* | |
| | | Test 3 | | *A. baumannii* | |
| C14 | | Test 1 | | *A. baumannii* | |
| | | Test 2 | | *A. baumannii* | |
| | | Test 3 | | *A. baumannii* | |
| C16 | | Test 1 | | N | |
| | | Test 2 | | N | |
| C18 | | Test 1 | | *A. baumannii* | |
| | | Test 2 | | *A. baumannii* | |
| | | Test 3 | | *A. baumannii* | |

*Appendix 1—table 5 Continued on next page*

Appendix 1—table 5 Continued

| Patient ID | Sample ID | Test No. | SSBD Results | CCT Results | NGS Results |
|---|---|---|---|---|---|
| C20 | | Test 1 | | N | |
| | | Test 2 | | *A. baumannii* | |
| | | Test 3 | | *A. baumannii* | |
| C22 | | Test 1 | | N | |
| | | Test 2 | | N | |
| | | Test 3 | | *A. baumannii* | |
| C24 | | Test 1 | | N | |
| | | Test 2 | | N | |
| C26 | | Test 1 | | *A. baumannii* | |
| | | Test 2 | | *A. baumannii* | |
| | | Test 3 | | *A. baumannii* | |
| C28 | | Test 1 | | *S. aureus* | |
| | | Test 2 | | N | |
| | | Test 3 | | *A. baumannii* | |
| C30 | | Test 1 | | N | |
| | | Test 2 | | *A. baumannii* | |
| | | Test 3 | | *A. baumannii* | |
| C32 | | Test 1 | | N | |
| | | Test 2 | | *K. pneumoniae* | |
| | | Test 3 | | *A. baumannii* *K. pneumoniae* | |
| C34 | | Test 1 | | N | |
| C36 | | Test 1 | | *S. aureus* | |
| | | Test 2 | | *A. baumannii* | |
| | | Test 3 | | *A. baumannii* | |
| C38 | | Test 1 | | *A. baumannii* | |
| | | Test 2 | | *A. baumannii* | |
| | | Test 3 | | *A. baumannii* | |
| C40 | | Test 1 | | *P. aeruginosa* | |
| | | Test 2 | | *P. aeruginosa* | |
| C42 | | Test 1 | | N | |
| | | Test 2 | | N | |
| C44 | | Test 1 | | *S. aureus* | |
| | | Test 2 | | *A. baumannii* | |
| C46 | | Test 1 | | N | |
| | | Test 2 | | N | |
| | | Test 3 | | *A. baumannii* | |
| C48 | | Test 1 | | *A. baumannii* | |
| | | Test 2 | | *A. baumannii* | |

**Appendix 1—table 6.** Comparative analysis of test results by SSBD, CCT and NGS in the validation stage II.

| Patient ID | Sample ID | Test No. | SSBD Results | CCT Results | NGS Results |
|---|---|---|---|---|---|
| B01 | B01-2 | Test 2 | N | N | N |
| B03 | B03-1 | Test 1 | N | N | N |
| B05 | B05-1 | Test 1 | *A. baumannii* (III) *S. maltophilia* (II) | *A. baumannii* | *A. baumannii* *S. maltophilia* |
| B07 | B07-1 | Test 1 | *P. aeruginosa* (III) | *P. aeruginosa* | *P. aeruginosa* |
| B11 | B11-1 | Test 1 | *A. baumannii* (III) *P. aeruginosa* (III) *S. maltophilia* (II) | *A. baumannii* | *A. baumannii* *P. aeruginosa* *S. maltophilia* |
| B13 | B13-1 | Test 1 | *S. epidermidis* (I) | N | N |
| B27 | B27-1 | Test 1 | *S. maltophilia* (I) | N | N |
| B29 | B29-1 | Test 1 | *K. pneumoniae* (III) | *K. pneumoniae* | *K. pneumoniae* |
| B31 | B31-1 | Test 1 | *A. baumannii* (III) *S. capitis* (III) | *A. baumannii* | *A. baumannii* *S. capitis* |
| B35 | B35-1 | Test 1 | *A. baumannii* (III) | *A. baumannii* | *A. baumannii* |
| B37 | B37-1 | Test 1 | N | N | *A. baumannii* |
| B41 | B41-1 | Test 1 | N | N | N |

**Appendix 1—table 7.** Antibiotic use of the patients prior to clinical trial in the experimental group and control group.

| | Empirical antibiotic therapy |
|---|---|
| **Experimental group** | |
| B01 | Piperacillin -tazobactam |
| B03 | Biapenem |
| B05 | Biapenem, Teicoplanin and Tigecycline |
| B07 | Biapenem |
| B09 | Biapenem and vancomycin |
| B11 | Cefoperazone-sulbactam and Tigecycline |
| B13 | Piperacillin-Tazobactam, trimethoprim-sulfamethoxazole and Teicoplanin |
| B15 | Imipenem-cilastatin and Linezolid |
| B17 | Biapenem |
| B19 | Biapenem |
| B21 | Ceftazidine-avibatam |
| B23 | Imipenem-Cilastatin |
| B25 | Imipenem-Cilastatin and Linezolid |
| B27 | Piperacillin-Tazobactam and trimethoprim-sulfamethoxazole |
| B29 | Imipenem-Cilastatin |
| B31 | Imipenem-Cilastatin and Teicoplanin |

*Appendix 1—table 7 Continued on next page*

*Appendix 1—table 7 Continued*

| | Empirical antibiotic therapy |
|---|---|
| B33 | Piperacillin-tazobactam |
| B35 | Moxifloxacin and Piperacillin-tazobactam |
| B37 | Biapenem, trimethoprim-sulfamethoxazole and Linezolid |
| B39 | Piperacillin-tazobactam |
| B41 | Piperacillin-tazobactam |
| B43 | Meropenem and Linezolid |
| Control group | |
| C02 | Meropenem and vancomycin |
| C04 | Biapenem and vancomycin |
| C06 | Piperacillin-tazobactam |
| C08 | Imipenem-Cilastatin |
| C10 | Cefoperazone-sulbactam |
| C12 | Moxifloxacin |
| C14 | Biapenem and Linezolid |
| C16 | Cefoperazone-sulbactam |
| C18 | Moxifloxacin |
| C20 | Piperacillin-tazobactam |
| C22 | Piperacillin-tazobactam and vancomycin |
| C24 | Cefoperazone-sulbactam, Linezolid and trimethoprim-sulfamethoxazole |
| C26 | Cefoperazone-sulbactam and Moxifloxacin |
| C28 | Piperacillin-tazobactam |
| C30 | Piperacillin-tazobactam |
| C32 | Piperacillin-tazobactam and Moxifloxacin |
| C34 | Meropenem |
| C36 | Cefoperazone-sulbactam |
| C38 | Piperacillin-tazobactam |
| C40 | Imipenem-Cilastatin |
| C42 | Cefoperazone-sulbactam |
| C44 | Piperacillin-tazobactam |
| C46 | Cefoperazone-sulbactam |
| C48 | Piperacillin-tazobactam and Linezolid |

**Appendix 1—table 8.** Patients' clinical outcomes.

| | Experimental group (n=22) | Control group (n=24) | p |
|---|---|---|---|
| Number of patients who have clinical indexes improved | | | |

*Appendix 1—table 8 Continued on next page*

*Appendix 1—table 8 Continued*

| | Experimental group (n=22) | Control group (n=24) | p |
|---|---|---|---|
| Day 3 vs. Day 1 | | | |
| Temperature, °C | 15 (68.2%) | 9 (37.5%) | 0.045* |
| WBC, $10^9$ /L | 15 (68.2%) | 12 (50.0%) | 0.211 |
| PCT, ng/mL | 18 (81.8%) | 19 (82.6%) | 0.945 |
| Day 7 vs. Day 1 | | | |
| Temperature, °C | 13 (72.2%) | 12 (54.5%) | 0.332 |
| WBC, $10^9$ /L | 16 (84.2%) | 11 (50.0%) | 0.021* |
| PCT, ng/mL | 13 (68.4%) | 19 (86.4%) | 0.166 |
| Day 10 vs. Day 1 | | | |
| Temperature, °C | 13 (82.6%) | 12 (70.6%) | 0.688 |
| WBC, $10^9$ /L | 9 (56.3%) | 8 (47.1%) | 0.598 |
| PCT, ng/mL | 12 (75.0%) | 15 (88.2%) | 0.325 |
| Number of patients undergoing effective treatment | | | |
| Day 3 | 13 (59.1%) | 11 (45.8%) | 0.395 |
| Day 7 | 16 (84.2%) | 11 (50.0%) | 0.046* |
| Day 10 | 13 (81.3%) | 10 (58.8%) | 0.259 |
| Clinical endpoint outcomes | | | |
| Hospital stay duration, days | 21 (13.7) | 23.5 (17.2) | 0.987 |
| 28 days mortality | 8 (36.4%) | 8 (33.3%) | 1.000 |
| Mechanical ventilation from randomization to 28th day, days | 11.3 (7.7) | 11.5 (7.7) | 0.970 |
| Shock from randomization to 28th day, days | 3.1 (4.3) | 2.3 (3.6) | 0.456 |
| Numbers of antibiotic-associated diarrhea | 0 (0.0%) | 2 (8.3%) | 0.490 |

For those data are n (%), all p values are calculated using Fisher's exact tests. For those data are mean (SD), all p values are calculated using Mann-Whitney tests. * indicated *P*-value <0.05.

**Appendix 1—table 9.** Potential competitive analysis among bacteria.

| Sample ID | Bacteria detected by SSBD grow in CCT tests | Bacteria detected by SSBD could not grow in CCT tests | Probable relations among bacteria |
|---|---|---|---|
| A05 | *K. pneumoniae* (III) *P. aeruginosa* (III) | *A. baumannii* (III) | *K. pneumoniae + P. aeruginosa > A. baumannii* |
| A09 | *A. baumannii* (II) | *S. aureus* (I) | Strength: II > I |
| A10 | *A. baumannii* (III) *K. pneumoniae* (III) | *P. aeruginosa* (I) | Strength: III + III > I |
| A11 | *A. baumannii* (III) | *K. pneumoniae* (I) | Strength: III > I |
| A16 | *K. pneumoniae* (III) | *P. aeruginosa* (III) | *K. pneumoniae > P. aeruginosa* |
| A21 | *A. baumannii* (III) *S. aureus* (I) | *K. pneumoniae* (III) | Strength: III + I > III |

*Appendix 1—table 9 Continued on next page*

*Appendix 1—table 9 Continued*

| Sample ID | Bacteria detected by SSBD grow in CCT tests | Bacteria detected by SSBD could not grow in CCT tests | Probable relations among bacteria |
|---|---|---|---|
| A22 | *A. baumannii* (III) | *S. epidermidis* (III) | *A. baumannii* > *S. epidermidis* |
| A27 | *A. baumannii* (III) | *S. epidermidis* (III) | *A. baumannii* > *S. epidermidis* |
| A28 | *A. baumannii* (II) | *P. aeruginosa* (III) | *A. baumannii* > *P. aeruginosa* |
| A30 | *A. baumannii* (III) | *S. aureus* (III) *S. epidermidis* (I) | *A. baumannii* > *S. aureus* + *S. epidermidis* |
| A33 | *S. aureus* (I) | *S. maltophilia* (II) | *S. aureus* > *S. maltophilia* |
| A36 | *A. baumannii* (III) | *S. maltophilia* (I) | *A. baumannii* > *S. maltophilia* |
| A37 | *A. baumannii* (III) | *S. epidermidis* (III) | *A. baumannii* > *S. epidermidis* |
| A45 | *A. baumannii* (III) | *K. pneumoniae* (II) | *A. baumannii* > *K. pneumoniae* |
| A49 | *A. baumannii* (III) *S. aureus* (III) | *S. maltophilia* (III) | *A. baumannii* + *S. aureus* > *S. maltophilia* |
| A55 | *A. baumannii* (III) | *E. coli* (II) *K. pneumoniae* (III) *S. maltophilia* (III) | *A. baumannii* > *E. coli. coli* + *K. pneumoniae* + *S. maltophilia* |
| A65 | *A. baumannii* (III) *P. aeruginosa* (I) | *S. maltophilia* (II) | Strength: III + I > II |
| A68 | *A. baumannii* (III) | *P. aeruginosa* (III) | *A. baumannii* > *P. aeruginosa* |
| A69 | *A. baumannii* (I) | *P. aeruginosa* (III) | *A. baumannii* > *P. aeruginosa* |
| A72 | *A. baumannii* (III) | *S. maltophilia* (I) | *A. baumannii* > *S. maltophilia* |
| A73 | *A. baumannii* (III) *P. aeruginosa* (III) | *K. pneumoniae* (III) | *A. baumannii* + *P. aeruginosa* > *K. pneumoniae* |
| A75 | *A. baumannii* (III) | *P. aeruginosa* (III) | *A. baumannii* > *P. aeruginosa* |
| A76 | *P. aeruginosa* (III) | *A. baumannii* (I) *S. maltophilia* (I) | Strength: III > I + I |
| B05-1 | *A. baumannii* (III) | *S. maltophilia* (II) | *A. baumannii* > *S. maltophilia* |
| B05-2 | *A. baumannii* (III) | *S. maltophilia* (I) | *A. baumannii* > *S. maltophilia* |
| B09-1 | *A. baumannii* (III) | *S. aureus* (III) *S. capitis* (III) *S. maltophilia* (III) | *A. baumannii* > *S. aureus* + *S. capitis* + *S. maltophilia* |
| B11-1 | *A. baumannii* (III) | *P. aeruginosa* (III) *S. maltophilia* (II) | *A. baumannii* > *P. aeruginosa* + *S. maltophilia* |
| B17-3 | *A. baumannii* (I) | *K. pneumoniae* (I) *E. faecalis* (I) | *A. baumannii* > *K. pneumoniae* +*E. faecalis* |
| B19-1 | *A. baumannii* (III) | *P. aeruginosa* (I) *S. maltophilia* (II) | Strength: III >I + II |
| B19-2 | *A. baumannii* (III), *S. maltophilia* (III) | *P. aeruginosa* (III) | *A. baumannii* + *S. maltophilia* > *P. aeruginosa* |
| B19-3 | *K. pneumoniae* (III), *S. maltophilia* (III) | *P. aeruginosa* (III) | *K. pneumoniae* + *S. maltophilia* > *P. aeruginosa* |
| B21-1 | *S. aureus* (III) *S. maltophilia* (III) | *P. aeruginosa* (III) | *S. maltophilia* + *S. aureus* > *P. aeruginosa* |

*Appendix 1—table 9 Continued on next page*

*Appendix 1—table 9 Continued*

| Sample ID | Bacteria detected by SSBD grow in CCT tests | Bacteria detected by SSBD could not grow in CCT tests | Probable relations among bacteria |
|---|---|---|---|
| B21-2 | *S. maltophilia* (III) | *S. aureu* (III), *P. aeruginosa* (III) | *S. maltophilia* > *S. aureus* + *P. aeruginosa* |
| B23-1 | *S. maltophilia* (III) | *S. capitis* (I) | Strength: III >I |
| B31-1 | *A. baumannii* (III) | *S. capitis* (III) | *A. baumannii* > *S. capitis* |
| B39-2 | *K. pneumoniae* (III) *P. aeruginosa* (III) | *A. baumannii* (I) *E. coli* (I) | Strength: III + III > I+I |

**Appendix 1—table 10.** Comparation of CCT, SSBD and NGS.

| | CCT | SSBD | NGS |
|---|---|---|---|
| Turnover time | 2–5 days | Less than 4 hours | 2–3 days |
| Cost | Low | Low | High |
| Detection target | Culturable bacteria | Selected targets | All microorganisms in the sample |
| Quantification | Semi | Relative | Semi |
| Instrument requirement | Low | Low | High |

