## [Editor Report]

Current culture-based, gold standard methods used for diagnosing the cause of sepsis provide results in 48-96 hours slowing antibiotic treatment initiation and leading to poor patient recovery. This work provides a new tool for identifying sepsis- and pneumonia-causing pathogens in less than 4 hours with species-specificity with the hope that the fast turnaround time leads to early treatment and improved clinical outcomes. Using an optimized PCR+CRISPR-Cas12a DNA detection method, the assay demonstrates good analytical sensitivity and specificity for 10 common bacterial pathogens that cause pneumonia. The method is validated in a clinical cohort and the clinical benefit is analyzed using a second cohort which is an intervention study used to guide clinicians on treatment choice.

---

## [Decision Letter]

**Decision letter after peer review:**

Thank you for submitting your article "Novel fast pathogen diagnosis method for severe pneumonia patients in the intensive care unit: randomized clinical trial" for consideration by *eLife*. Your article has been reviewed by 3 peer reviewers, and the evaluation has been overseen by a Reviewing Editor and Bavesh Kana as the Senior Editor. The reviewers have opted to remain anonymous.

Essential revisions:

1) It is not clear which epidemiological data these pathogens were selected from. Moreover, the selected pathogens are only bacteria. Can the authors expand?

2) The authors imply that the implementation of their method has a clear clinical benefit but the data supporting this claim is sparse. Although the method seemed to improve disease severity, no improvement was indicated for important clinical measures including organ function, time of ventilation, and mortality. Testing more samples may provide a better understanding of the clinical benefit of the method but the data presented provide limited evidence. Are further data available? Hospital stay duration, survival? If not, these statements should be reconsidered.

3) BALF is the primary sample used for diagnosis in this work but can be difficult to obtain in patients with severe symptoms and young children. As a result, it may not be as useful in these patient populations thus reducing the application of this method. Were other sample types tested? Can the authors comment on this limitation?

4) The advantage of SSBD over NGS is clear but the benefit of SSBD compared to tests such as BioFire and Curetis could be made clearer. It seems that these two methods show low clinical specificity and thus cannot be used in the ICU but data or further explanation of this claim would validate the advantage of SSBD to current sepsis detection methods.

5) Consider creating a table for direct comparison of SSBD to CCT and NGS, rather than simply using 12 random samples for NGS.

6) Consider adding time taken to receive results for cultures, SSBD, and NGS to highlight the rapidity of the novel diagnostic test.

7) Consider investigating potential ways to distinguish polymicrobial results found on SSBD testing from colonization/non-pathogenic infections.

Other concerns that must be addressed:

1) Page 9. Please explain how primer specificity was evaluated.

2) Page 9. Were patients on antibiotics before getting into the trial? Add this detail.

3) Page 10 At which timepoint the patients received different treatment based on the results of the culture or SSBD? Was this consistent?

4) Page 11. The second sentence of 3.1 section in results is not clear.

5) How were patients allocated to groups? Randomised?

6) The table describing the patient cohort is in supplementary information. It should be in the main manuscript. It seems that the control and experimental groups were not balanced. Please explain.

7) What was the threshold level of fluorescence (in Figure 3) which was considered important?

*Reviewer #1 (Recommendations for the authors):*

Did the experimental groups have any differences in hospital stay, or survival? It would be nice to show if there is a correlation with important clinical outcomes.

*Reviewer #2 (Recommendations for the authors):*

The advantage of SSBD over NGS is clear but the benefit of SSBD compared to BioFire and Curetis could be improved. It seems that these two methods show low clinical specificity and thus cannot be used in the ICU but data or further explanation of this claim would further validate the advantage of SSBD to current sepsis detection methods.

*Reviewer #3 (Recommendations for the authors):*

Consider creating a table for direct comparison of SSBD to CCT and NGS rather than simply 12 random samples for NGS.

– Consider adding time until results received for cultures, SSBD, and NGS to highlight the rapidity of the novel diagnostic test.

– Consider investigating potential ways to distinguish polymicrobial results found on SSBD testing from colonization/non-pathogenic infections?

---

## [Author Response]

Essential revisions:(1) It is not clear which epidemiological data these pathogens were selected from. Moreover, the selected pathogens are only bacteria. Can the authors expand?

Thank you for the question and sorry for the unclear explanation.

Here we added the reason in Methods part section 2.3 (page 8, line 160-162) and labeled as red. Briefly, based on the pathogenic bacteria record of China and worldwide in both previous studies (Zhou et al., 2014. *PLoS One*; Sakr et al., 2018. *Open Forum Infectious Diseases*) and local record (2017, appendix 1-figure 2), the selected 10 species covered 76% sepsis pathogenic bacteria in ICU.

In ICU, many microbes could cause sepsis other than bacteria, such as fungus, viruses, mycoplasma. However, bacteria belong to the major pathogenic microbes that infected more than 84% patients (Sakr et al., 2018. *Open Forum Infectious Diseases*). On the other hand, according to the high similarity and huge differences in medication decision, identifying pathogenic bacteria is the most difficult but important task among all pathogenic microbes (Rhodes et al., 2017. *Intensive Care Medicine*). Based on these two aspects, we started this study from bacteria.

The new tool SSBD was based on specie-specific target of pathogenic microbes, which means it can expend to any species other than bacteria. The identification targets could also expend to common drug-resistant genes to provide further information for clinical decision. We have already started to launch a multicenter clinical trial to further evaluate this new tool, which fungus and drug resistant genes were added to the diagnosis targets.

(2) The authors imply that the implementation of their method has a clear clinical benefit but the data supporting this claim is sparse. Although the method seemed to improve disease severity, no improvement was indicated for important clinical measures including organ function, time of ventilation, and mortality. Testing more samples may provide a better understanding of the clinical benefit of the method but the data presented provide limited evidence. Are further data available? Hospital stay duration, survival? If not, these statements should be reconsidered.

We appreciate the comments and the questions. In this study, we planned to evaluate the accuracy and the clinical benefit of the new method SSBD. For the first purpose, we did see the high sensitivity and specificity of SSBD in bacteria detection. For the second purpose, the SSBD significantly shorten the turnover time of obtaining the pathogenic information and definitely help the clinical doctors to make better decisions. In addition, the experimental group demonstrated faster antibiotic coverage and better APACHE II scores comparing with control group. Based on this information, we drew the conclusion that SSBD was an accurate tool with great potential but need “to be applied in more clinical research” (page 21, line 449-450).

As suggested by the reviewer, we added the discussion about the important clinical measures in the discussion (page 20, line 417-419). As showed in appendix table 7, patients in the experimental group for example demonstrated significant better measures of temperature improvement at day 3, WBC improvement at day 7. However, other measures did not demonstrate significant difference between patients from two groups. As we mentioned in the discussion part, the samples size (22 vs 24), potential drug-resistant genes, other pathogenic microbes other than 10 selected bacteria may limit the clinical benefit of SSBD.

To better evaluate SSBD, we have already started to launch a multi-center clinical trial to further evaluate this new tool. Hopefully we could have a better answer to reviewer’s concerns when we finished this trial with better design and larger size.

(3) BALF is the primary sample used for diagnosis in this work but can be difficult to obtain in patients with severe symptoms and young children. As a result, it may not be as useful in these patient populations thus reducing the application of this method. Were other sample types tested? Can the authors comment on this limitation?

We appreciate your suggestions and questions.

First, please allow us to explain why we chose BALF as samples in this study. Based on the previous studies and international guidance, BALF is the best sample which contains less contamination and non-colonized bacteria (Torres et al., 2017. *European Respiratory Journal*). Considering the real-world situation of patients in ICU, all participants were older than 18 and applied artificial airway, which means the acquisition of BALF is general operation with no difficulty.

On the other hand, we highly agree with reviewer’s opinion that various types of samples would expend the value of SSBD. Based on the principle, any samples with enough DNA information could be test samples for SSBD. In our preliminary tests and other studies, we successfully acquired accurate pathogenic microbe information from sputum, blood, amniotic fluid, cerebrospinal fluid, pus and tissues. Among them, extra sample preparation processes were needed for sputum due to its viscous status. Multiple mature methods were reported in previous studies (Kim et al., 2014. *Annals of Laboratory Medicine*; Stokell et al., 2014. *Journal of Clinical Microbiology*).

(4) The advantage of SSBD over NGS is clear but the benefit of SSBD compared to tests such as BioFire and Curetis could be made clearer. It seems that these two methods show low clinical specificity and thus cannot be used in the ICU but data or further explanation of this claim would validate the advantage of SSBD to current sepsis detection methods.

Thank you for the suggestions.

Since the diagnosis products from BioFire and Curetis were neither approved by CFDA nor widely used in China, we had no access to comparing these two methods head-to-head in our study. Based on our understanding, both methods were approved by FDA and used in hospitals in the United States. According to their product instruction and clinical trial results, FilmArray Pneumonia Panel (BioFire) demonstrated various false positive results in bacteria detection.

**Author response image 1. sa2fig1:** 

Based on the open information about the primers and probes, such false positive results may due to the highly similarities in the selected gene regions. This bottleneck was solved in SSBD due to better DNA markers. To better describe this information, we rewrote relative part in the revised manuscript (page 17, line 359-361).

(5) Consider creating a table for direct comparison of SSBD to CCT and NGS, rather than simply using 12 random samples for NGS.

Thank you for the suggestions.

We extracted all the sample results with all three tests: SSBD, CCT and NGS, and listed as appendix 1-table 5 for stage I (11 samples) and appendix 1-table 6 for stage II (12 samples). We are sorry for the misunderstanding expression about “random samples” in the last version of manuscript. In our clinical trial design, NGS was not included as a mandatory test. Therefore, such information was only acquired when patients determined to get extra information by NGS, which was neither a requirement of current guidance nor covered by medical insurance in China. Luckily, we had 23 samples with NGS results in two stages, and we think a comparison among three methods would be interesting to demonstrate in the manuscript.

(6) Consider adding time taken to receive results for cultures, SSBD, and NGS to highlight the rapidity of the novel diagnostic test.

Thank you for the great suggestion. We added an appendix 1-table 10 to better compare SSBD, CCT and NGS from several main aspects, including the most important one “turnover time” (page 31, Appendix 1)

(7) Consider investigating potential ways to distinguish polymicrobial results found on SSBD testing from colonization/non-pathogenic infections.

We highly appreciate this suggestion.

In fact, this is a problem that has puzzled clinical doctors for a long time. Such distinguish of colonization/non-colonization is a challenge for CCT and NGS. One of the most important reason we chose BALF as sample was to try the best to avoid the effect of colonization. Based on the previous studies, most nonpathogenic bacteria might be eliminated during the acquisition of BALF (Fagon et al., 2000. *Annals of Internal Medicine*). On the other hand, it is more important to determine the antibiotic that covers all pathogenic bacteria of sepsis patient in ICU rather than miss some of them, which means nonpathogenic infection was not the top priority issue.

Recently, several studies provided some promising solutions to this problem (Dickson et al., 2014. *Lancet Respiratory Medicine*; Pettigrew et al., 2020).

*Journal of Infectious Diseases*. In our study, we are trying to target mRNA fragments instead of genome DNA to distinguish colonization and noncolonization by different gene expression patterns for some critical bacteria in sepsis treatment. Hopefully we can demonstrate some promising results in our next publication.

Other concerns that must be addressed:(1) Page 9. Please explain how primer specificity was evaluated.

Thank you for the suggestions and sorry for the unclear expression. As we mentioned in the answer to question 4, the diagnosis products from BioFire and Curetis were neither approved by CFDA nor widely used in China, we had no access to test the primers on bench. Instead, we selected one example based on the open information, and evaluated the alignment of DNA regions amplified by primers to multiple reference genomes as described in Method part section 2.3 (page 9, line 170-179).

For the primers we designed in SSBD, similar process was applied and no unspecific target was discovered in the representative genome from *S. epidermidis*. Besides, the primers in SSBD were further tested with different clinical strains of target species and cross validated (page 13, line 265-268, Figure 3C).

(2) Page 9. Were patients on antibiotics before getting into the trial? Add this detail.

Thank you for the question. An appendix 1-table 7 was added to demonstrate this information (page 24-25, Appendix 1).

(3) Page 10 At which timepoint the patients received different treatment based on the results of the culture or SSBD? Was this consistent?

Thank you for the question.

Based on the clinical design, patients from both groups would receive treatment every day during the study, which are consistent in experimental and control groups. Besides the clinical measures, clinical doctors have timely extra pathogenic information about the patients in experimental group to make better treatment decision due to shorter turnover time of SSBD. For example, at day 1, SSBD reported the pathogenic information in 4 hours, while CCT reported the pathogenic information after 2 days.

(4) Page 11. The second sentence of 3.1 section in results is not clear.

Thank you for the suggestion and sorry for the unclear expression. We rewrote this sentence and provided some examples (page 11, line 227231).

(5) How were patients allocated to groups? Randomised?

Sorry for the inconspicuous expression.

The randomized allocation of patients to two groups was introduced in section 2.4 “patients” in the manuscript (page 10, line 188-190) and detailly described in the appendix file section “Sample size and randomized double-blind trial” (page 2, line 28-32, Appendix 1).

(6) The table describing the patient cohort is in supplementary information. It should be in the main manuscript. It seems that the control and experimental groups were not balanced. Please explain.

Thank you for the important suggestions.

We have moved the patient cohort information table to the main table part as table 1. We are not sure what does the reviewer mean “not balanced”. If it refers to the patient number (22 vs 24), it was due to the randomized process (see line 28-32 in appendix 1). If it refers to the baseline characteristics, only age demonstrated significant difference between control and experimental groups (68 ±9.5 vs 58±17.4), which might be due to the limit size of patients. All other baseline characteristics had no significant difference between two groups.

(7) What was the threshold level of fluorescence (in Figure 3) which was considered important?

We appreciate the question and sorry for the unclear expression.

In figure 3, which aimed to test whether SSBD could identify the target bacteria, the positive threshold level of fluorescence was set to be significantly higher than negative control. Negative control stood for the fluorescence values of PCR products of using DEPC-H2O as input, and the significance was estimated using an unpaired t-test.

Later on, in the clinical study, a more detailed standard was applied to further determine the relative amount of target bacteria (appendix 1, page 4, line 7180)